# Autistic traits, but not schizotypy, predict increased weighting of sensory information in Bayesian visual integration

**Povilas Karvelis[1], Aaron R Seitz[2], Stephen M Lawrie[3,4], Peggy Seriès[1]***

[1]IANC, School of Informatics, University of Edinburgh, Edinburgh, United Kingdom; [2]Department of Psychology, UC Riverside, Riverside, United States; [3]Division of Psychiatry, University of Edinburgh, Edinburgh, United Kingdom; [4]Patrick Wild Centre, University of Edinburgh, Edinburgh, United Kingdom

**Abstract** Recent theories propose that schizophrenia/schizotypy and autistic spectrum disorder are related to impairments in Bayesian inference that is, how the brain integrates sensory information (likelihoods) with prior knowledge. However existing accounts fail to clarify: (i) how proposed theories differ in accounts of ASD vs. schizophrenia and (ii) whether the impairments result from weaker priors or enhanced likelihoods. Here, we directly address these issues by characterizing how 91 healthy participants, scored for autistic and schizotypal traits, implicitly learned and combined priors with sensory information. This was accomplished through a visual statistical learning paradigm designed to quantitatively assess variations in individuals' likelihoods and priors. The acquisition of the priors was found to be intact along both traits spectra. However, autistic traits were associated with more veridical perception and weaker influence of expectations. Bayesian modeling revealed that this was due, not to weaker prior expectations, but to more precise sensory representations.
DOI: https://doi.org/10.7554/eLife.34115.001

**\*For correspondence:**
pseries@inf.ed.ac.uk

**Competing interests:** The authors declare that no competing interests exist.

## Introduction

In recent years Bayesian inference has come to be regarded as a general principle of brain function that underlies not only perception and motor execution, but hierarchically extends all the way to higher cognitive phenomena, such as belief formation and social cognition. Impairments of Bayesian inference have been proposed to underlie deficits observed in mental illness, particularly schizophrenia (*Fletcher and Frith, 2009*; *Corlett et al., 2009*; *Adams et al., 2013*; *Hemsley and Garety, 1986*; *Friston, 2005*; *Stephan et al., 2006*) and autistic spectrum disorder (ASD) (*Pellicano and Burr, 2012a*; *Van de Cruys et al., 2014*; *Lawson et al., 2014*; *Palmer et al., 2017*). The general hypothesis for both disorders is that the weight, also called 'precision', ascribed to sensory evidence and prior expectations is imbalanced, resulting in sensory evidence having relatively too much influence on perception.

In schizophrenia, overweighting of sensory information could explain the decreased susceptibility to perceptual illusions (*Notredame et al., 2014*), as well as the peculiar tendency to jump to conclusions (*Speechley et al., 2010*). Moreover, the systematically weakened low-level prior expectations might lead to forming compensatory strong and idiosyncratic high-level priors (beliefs), which would explain the emergence and persistence of delusions as well as reoccurring hallucinations (*Fletcher and Frith, 2009*; *Corlett et al., 2009*; *Adams et al., 2013*).

In ASD, the relatively stronger influence of sensory information could explain hypersensitivity to sensory stimuli and extreme attention to details. The weaker influence of prior expectations would also result in more variability in sensory experiences. The desire for sameness and rigid behaviors

could then be understood as an attempt to introduce more predictability in one's environment (*Pellicano and Burr, 2012a*). Furthermore, this could lead to prior expectations which are too specific and which do not generalize across situations (*Van de Cruys et al., 2014*). While all theories agree that the relative influence of prior expectations is weaker in ASD, the primary source of this imbalance is debated: does it arise from increased sensory precision (i.e. sharper likelihood) or from reduced precision of prior expectations? (*Brock, 2012*; *Pellicano and Burr, 2012b*; *van Boxtel and Lu, 2013*) (*Figure 1*). Some authors argue for attenuated priors (*Pellicano and Burr, 2012a*; *Pellicano and Burr, 2012b*), while others argue for increased sensory precision (*Lawson et al., 2014*; *Palmer et al., 2017*; *Brock, 2012*; *Van de Cruys et al., 2013*) but conclusive experimental evidence is lacking.

A number of studies have aimed at testing Bayesian theories, either in a clinical population, or by studying individual differences in the general population (*Powell et al., 2016*; *Skewes et al., 2015*; *Teufel et al., 2015*; *Schmack et al., 2013*) under the hypothesis of a continuum between autistic/ schizotypal traits and ASD/schizophrenia (*Nelson et al., 2013*; *van Os et al., 2009*; *Constantino and Todd, 2003*).

Attenuated slow-speed priors were reported in a motion perception task in individuals with ASD traits (*Powell et al., 2016*). Autistic children also showed attenuated central tendency prior in temporal interval reproduction (*Karaminis et al., 2016*). Attenuated priors were also reported in perceptual tasks that incorporate probabilistic reasoning (*Skewes et al., 2015*; *Skewes and Gebauer, 2016*). However, the direction of gaze priors (*Pell et al., 2016*) and the light-from-above priors (*Croydon et al., 2017*) were found to be intact. Autistic children also demonstrated intact ability to update their priors in a volatile environment in a decision-making task (*Manning et al., 2017*) but a follow-up study in ASD adults showed that they overestimate volatility in a changing environment (*Lawson et al., 2017*).

In schizophrenia/schizotypal traits, *Teufel et al. (2015)* reported increased influence of prior expectations when disambiguating two-tone images, while *Schmack et al. (2015*, *2017)* reported weakened influence of stabilizing predictions when observing a bistable rotating sphere.

Overall, the existing findings are not only mixed, but also employ very different paradigms, which makes their direct comparison difficult. Further, a critical limitation of most studies (except for *Karaminis et al., 2016*) is the lack of formal computational models that can test whether behavioral differences originate from different priors or from different likelihoods. Moreover, to our knowledge, despite the similarity of the Bayesian theories proposed for ASD and schizophrenia, there is no

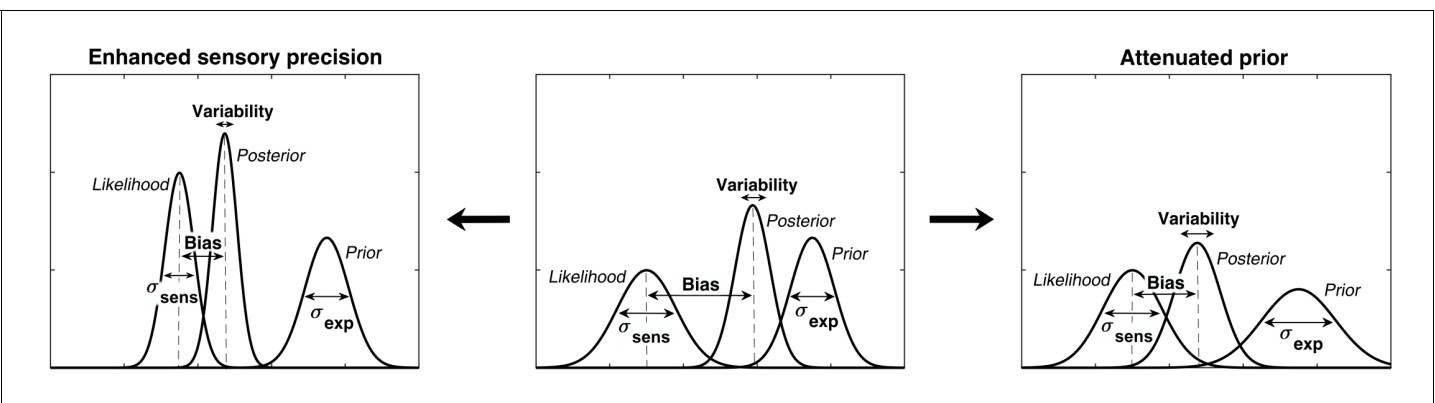

**Figure 1.** Alternative hypotheses for ASD impairments within the Bayesian inference framework. In Bayesian terms, the percept can be described as a posterior distribution, which is a combination of sensory information (likelihood) and prior expectations (prior). Two contrasting hypotheses have been proposed to underlie behavioral differences in ASD: enhanced sensory precision, that is, smaller $\sigma_{sens}$ (left) vs. attenuated priors, that is, larger $\sigma_{exp}$ (right). Both hypotheses predict a reduced influence (bias) of the prior on the location of the posterior distribution (posterior mean). However, these alternatives differ in their predictions for perceptual variability, which is determined by the posterior width: the enhanced sensory precision hypothesis should lead to reduced variability while the attenuated prior hypothesis should lead to increased variability. By measuring both bias and variability, our experimental paradigm can distinguish between these two hypotheses.
DOI: https://doi.org/10.7554/eLife.34115.002

previous work investigating both autistic and schizotypal traits within the same experimental paradigm so as to test their differences.

We here address these questions empirically in a context of visual motion perception. We used a previously developed statistical learning task (*Chalk et al., 2010*) in which participants have to estimate the direction of motion of coherently moving clouds of dots (*Figure 2*). *Chalk et al. (2010)* found that in this task healthy participants rapidly and implicitly develop prior expectations for the most frequently presented motion directions. This in turn alters their perception of motion on low contrast trials resulting in attractive estimation biases towards the most frequent directions. In addition, prior expectations lead to reduced estimation variability and reaction times, as well as increased detection performance for the most frequently presented directions. When no stimulus is presented, the acquired expectations sometimes lead to false alarms (hallucinations), again, mostly in the most frequent directions. Importantly, such biases were well described using a Bayesian model, where participants acquired a perceptual prior for the visual stimulus that is combined with sensory information and influences their perception. As such, this paradigm is well suited to quantitatively model variations in likelihoods and priors in individuals with ASD or schizotypal traits.

## Results

Here, we investigated individual differences in statistical learning in relation to autistic and schizotypal traits in a sample of 91 healthy participants. Eight participants failed to perform the task satisfactorily and were excluded from the analysis (see Materials and methods), leaving 83 participants in the study (41 women and 42 men, age range: 18–69; mean: 25.7).

### Task behavior at low contrast

First, we investigated whether participants acquired priors on the group level. We discarded the first 170 trials as that is how long it took for the 2/1 and 4/1 staircases contrast levels to converge (*Appendix 1—figure 2*) and for prior effects to become significant (*Appendix 1—figures 3*, *4* and *5*). We analyzed task performance at low contrast levels (converged 2/1 and 4/1 staircases contrast levels) where sensory uncertainty is high. Replicating findings of *Chalk et al. (2010)*, we found that on the group level people acquired priors that approximated the statistics of the task. Such priors

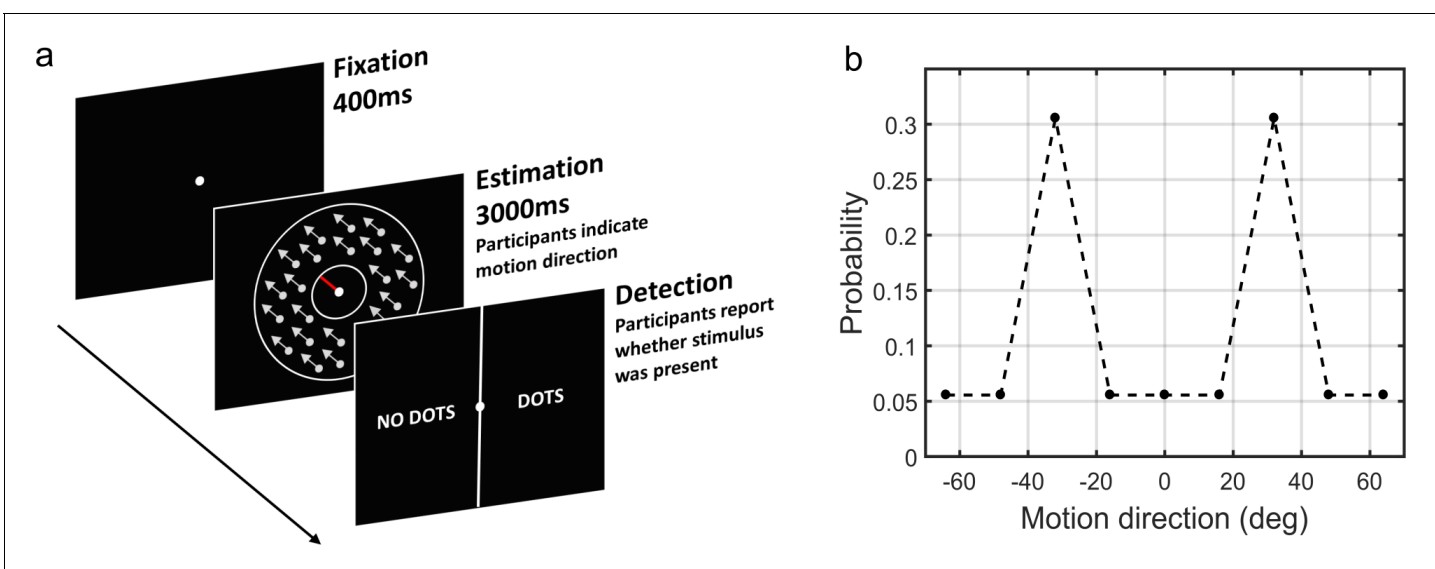

**Figure 2.** The moving dots task. (a) Sequence of events on a single trial. First, a fixation point is presented. Next, a field of coherently moving dots is presented along with an estimation bar (extending from the fixation point) which participants are required to move to indicate perceived motion direction. Lastly, in a two-alternative forced choice, participants are asked to report whether they saw the dots during the estimation part (detection task). (b) The probability of different motion directions being presented: directions at ±32° are presented more often than other directions. Motion direction is plotted relative to a central reference angle (at 0°), which was randomly set for each participant.
DOI: https://doi.org/10.7554/eLife.34115.003

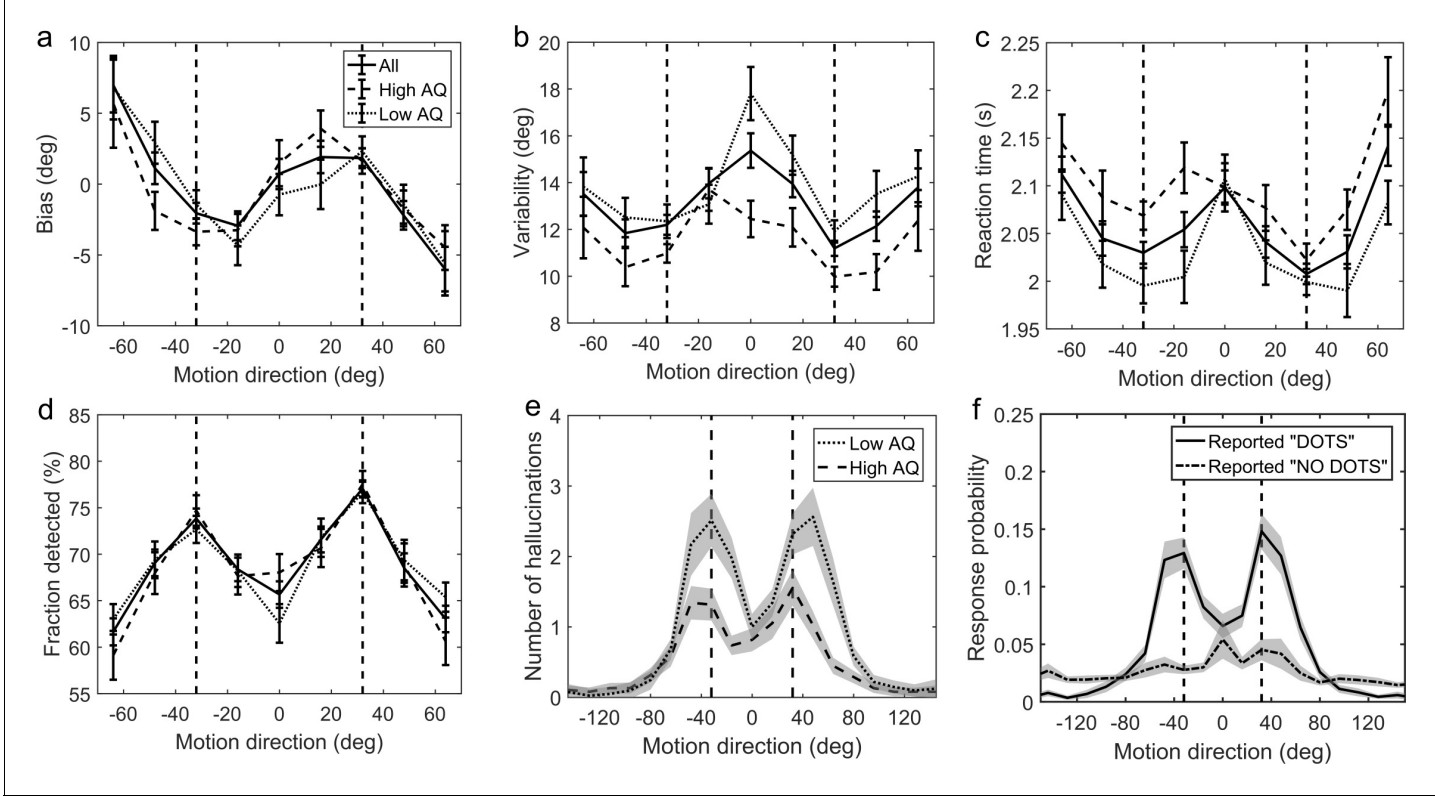

**Figure 3.** Average group performance on low-contrast trials (**a–d**) and on trials with no stimulus (**e**). (**a**) Mean estimation bias, (**b**) standard deviation of estimations, (**c**) estimation reaction time and (**d**) fraction of trials in which the stimulus was detected. (**f**) Probability distribution of estimation responses on trials without stimulus. The solid line denotes the estimation responses when participants reported detecting a stimulus (hallucinations). The dash-dot line denotes estimation distributions when participants correctly reported not detecting a stimulus. (**e**) Distribution of hallucinations for high and low AQ groups (median split). The vertical dashed lines correspond to the two most frequently presented motion directions (±32°). Error bars and shaded areas represent within-subject standard error.

DOI: https://doi.org/10.7554/eLife.34115.004

The following source data is available for figure 3:

**Source data 1.** This zip archive contains .csv files with all of the data that was used to produce plots in *Figure 3*.

DOI: https://doi.org/10.7554/eLife.34115.005

were indicated by: attractive biases towards ±32° (*Figure 3a*), less variability in estimations at ±32° (*Figure 3b*; standard deviation of estimations 11.9 ± 0.30° at ±32° versus 13.84 ± 2.38° over all other motion directions; signed rank test: p<0.001), shorter estimation reaction times at ±32° as compared to all other motion directions (*Figure 3c*; average reaction time was 201.87 ± 2.47 ms at ±32° versus 207.75 ± 2.60 ms over all other motion directions; signed rank test: p<0.001) and better detection at ±32° as compared to all other motion directions (*Figure 3d*; detected 75.57 ± 0.65% at±32° versus 66.70 ± 0.83% over all other motion directions; signed rank test: p<0.001).

## No-stimulus performance

Another indicator of acquired priors is the distribution of estimation responses on trials when no actual stimulus was presented. We found that participants sometimes still reported seeing dots (experienced hallucinations) but mostly so around ±32° (*Figure 3f*, solid line). To quantify the statistical significance of hallucinations around ±32°, the space of possible motion directions was divided into 45 bins of 16° and the probability of estimation within 8° of ±32° was multiplied by the total number of bins:

$$p_{rel} = p(\theta_{est} = \pm 32(\pm 8)^\circ) \cdot N_{bins}, \tag{1}$$

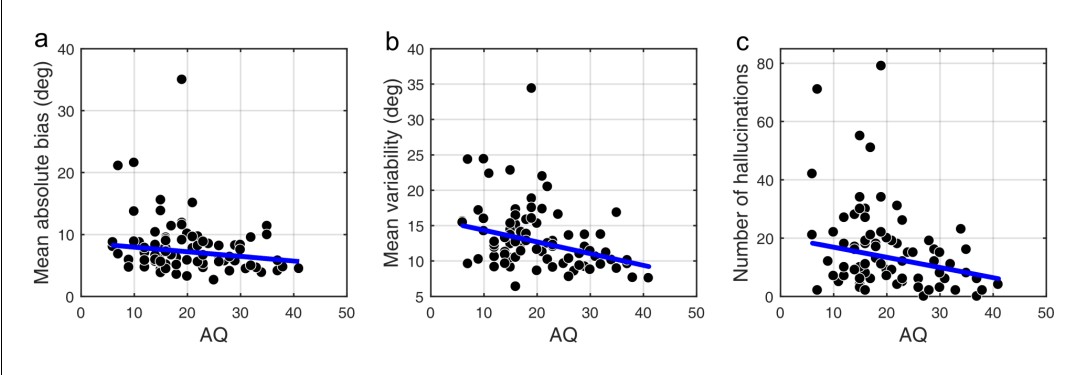

**Figure 4.** Correlations between AQ scores and task performance on low contrast trials (**a, b**) and when no stimulus is presented (**c**). (**a**) Mean absolute bias ($r = -0.175$, $p=0.053$), (**b**) mean standard deviation (i.e. variability) of estimations ($r = -0.327$, $p<0.001$, and (**c**) the total number of hallucinations ($r = -0.238$, $p=0.010$). The blue lines are robust regression slopes.
DOI: https://doi.org/10.7554/eLife.34115.006
The following source data is available for figure 4:

**Source data 1.** This zip archive contains .csv files with all of the data that was used to produce plots in *Figure 4*.
DOI: https://doi.org/10.7554/eLife.34115.007

where $N_{bins}$ is the number of bins (45), each of size 16°. This probability ratio would be equal to one if participants were equally likely to estimate within 8° of ±32°, as they were to estimate within other bins. We found that the median of $p_{rel}$ was significantly greater than 1 (median($p_{rel}$)=1.6, $p<0.001$, signed rank test). Furthermore, the estimation distribution when no dots where detected (*Figure 3f*, dash-dot line) was found to be significantly flatter (median($p_{rel}$)=0, $p<0.001$, signed rank test comparing with the median of $p_{rel}$ for hallucinations), suggesting that the hallucinations were indeed of perceptual nature (rather than related to a response bias).

## Task performance and autistic/schizotypy traits

Participants were prescreened to make sure they covered a wide range of autistic and schizotypy scores. The AQ scores in our sample ranged from 6 to 41 with a mean (±SD) of 20.3 (±8.3). The RISC scores ranged from 8 to 55 with a mean of 31.7 (±11.9), and the SPQ scores ranged from 4 to 59 with a mean of 26.4 (±13.8).

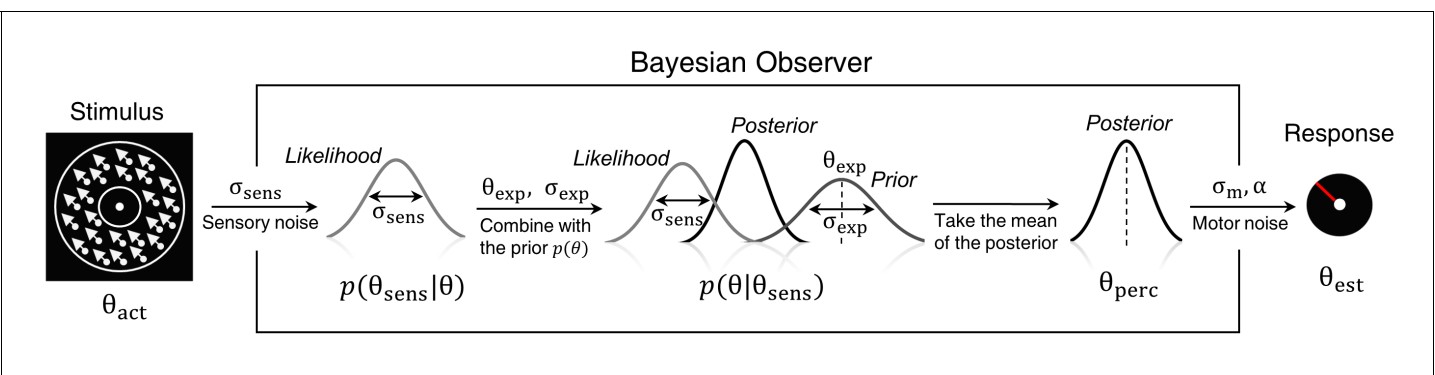

**Figure 5.** Bayesian model of estimation response for a single trial. The actual motion direction ($\theta_{act}$) is corrupted by sensory uncertainty ($\sigma_{sens}$), and then combined with prior expectations (mean $\theta_{exp}$ and uncertainty $\sigma_{exp}$) to form a posterior distribution. The perceptual estimate ($\theta_{perc}$) is defined as the mean of the posterior distribution. Finally, motor precision ($1/\sigma_m^2$) and a probability of random response ($\alpha$) are incorporated to generate the response ($\theta_{est}$). This results in four free model parameters: $\sigma_{sens}$, $\sigma_{exp}$, $\theta_{exp}$ and $\alpha$. The motor precision is estimated from high contrast trials and is used as a fixed parameter.
DOI: https://doi.org/10.7554/eLife.34115.008

We found that on low contrast trials autistic traits lead to less variability in estimations (*Figure 4b*; mean standard deviation of estimations: $r = -0.327$, p<0.001), which remained significant after Bonferroni correction (p=0.002). Moreover, there was a negative relationship between autistic traits and estimation bias, which was trending according to robust regression (*Figure 4a*; mean absolute estimation bias: $r = -0.175$, p=0.053) and significant according to Kendall's correlation ($\tau_b = -0.163$, p=0.032), however, it did not survive Bonferroni correction (p=0.212). In the Bayesian framework, less bias could arise either due to wider priors or narrower sensory likelihoods, while less variability could be a result of either narrower priors or narrower likelihoods (see *Figure 1*). Thus, observing less bias and less variability together suggests that the effects are driven by narrower likelihoods. An alternative is that the differences in variability could be due to differences in motor precision, which we further assess via modeling (below).

Schizotypy traits (RISC and SPQ scores) did not show any effect on task performance at low contrast as indicated by the absence of correlations with mean absolute estimation bias (RISC: $r = 0.140$, p=0.197; SPQ (N = 39): $r = -0.160$, p=0.204) and with mean estimation variability (RISC: $r = 0.197$, p=0.092; SPQ (N = 39): $r = -0.229$, p=0.171); see *Appendix 1—figures 6, 7* and *8*.

## No-stimulus trials and autistic/schizotypal traits

We also investigated how the traits affected performance on trials when no actual stimulus was presented. First, we looked at the total number of estimations. We found that autistic traits were associated with less hallucinations (*Figure 4c*; $r = -0.238$, p=0.010), while schizotypal traits were found to have no effect on the number of hallucinations (RISC: $r = 0.126$, p=0.163; SPQ (N = 39): $r = -0.010$, p=0.959). Secondly, we looked for relationships between the traits and how the estimations on no-stimulus trials were distributed. Specifically, we were interested in whether the traits predicted how

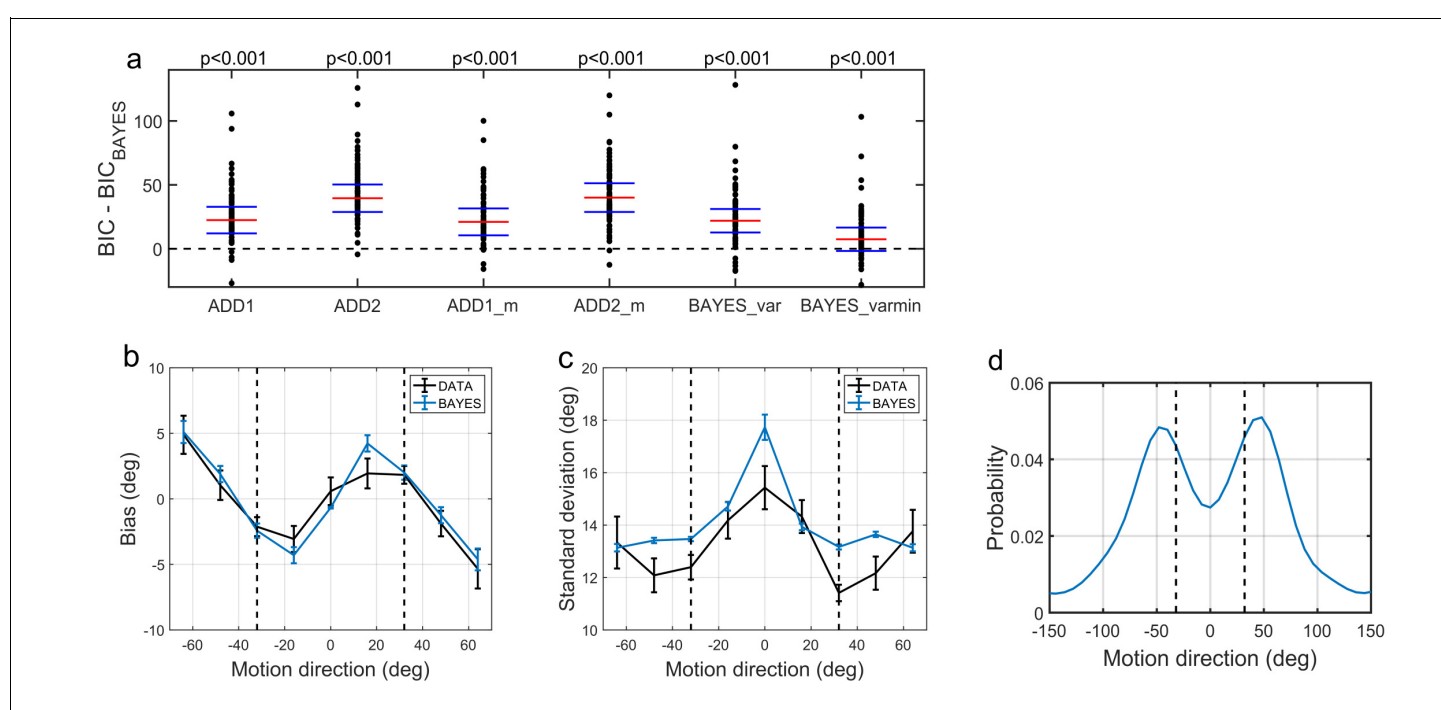

**Figure 6.** Modelling results. (a) Model comparison for all participants using Bayesian Information Criterion (BIC). y-axis measures the relative difference between BIC of each model (as indicated on the x-axis) and BIC of BAYES model. Values greater than zero on the y-axis indicate that the BAYES model provided a better fit. Each dot represents a participant. Red horizontal lines denote median values; blue horizontal lines denote 25th and 75th percentiles. p-values above the plot indicate whether the median of the difference was significantly different from zero for each model (signed rank test). Panels (a) and (c) present task performance at different motion directions as predicted by BAYES model: (b) estimation bias, (c) standard deviation of estimations. Error bars represent within-subject standard error. (d) Population averaged prior as recovered via BAYES model. The vertical dashed lines correspond to the two most frequently presented motion directions (±32°).
DOI: https://doi.org/10.7554/eLife.34115.009

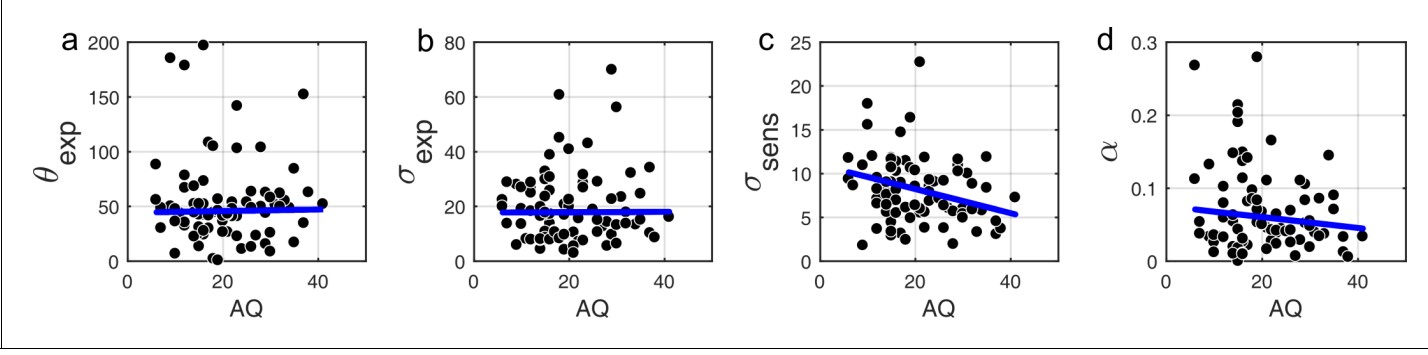

**Figure 7.** Correlations between AQ scores and BAYES model parameters. (a) $\theta_{exp}$ - mean of the prior expectations ($r$ = 0.031, p=0.820), (b) $\sigma_{exp}$ - uncertainty of the prior distribution ($r$ = 0.018, p=0.962), (c) $\sigma_{sens}$ - uncertainty in the sensory likelihood ($r$ = −0.185, p=0.011) and (d) $\alpha$ - fraction of random estimations (r = −0.135, p=0.238). The blue lines are robust regression slopes.
DOI: https://doi.org/10.7554/eLife.34115.010

The following source data is available for figure 7:

**Source data 1.** This zip archive contains .csv files with all of the data that was used to produce plots in *Figure 7*.
DOI: https://doi.org/10.7554/eLife.34115.011

densely hallucinations were distributed around ±32°, as this could be considered to reflect the differences in the width of the underlying acquired prior distribution. For weaker priors we would expect a more spread out distribution of hallucinations. To test this hypothesis, we looked at the fraction of total hallucinations in the region around ±32° for three different-sized windows: Within 8°, within 16° and within 24° of ±32°. Bayesian Kendall correlation analysis on these measures provided positive evidence that none of the traits had any effect on how hallucinations were distributed, suggesting no differences in the acquired prior distributions (fraction of hallucinations within 8° of ±32°: AQ - $\tau_b$ = 0.003, $BF_{01}$ = 7.24; RISC - $\tau_b$ = -0.050, $BF_{01}$ = 3.73; SPQ - $\tau_b$ = 0.101, $BF_{01}$ = 8.72; within 16° of ±32°: AQ - $\tau_b$ = -0.068, $BF_{01}$ = 2.86; RISC - $\tau_b$ = -0.129, $BF_{01}$ = 0.84; SPQ - $\tau_b$ = 0.018, $BF_{01}$ = 5.45; within 24° of ±32°: AQ - $\tau_b$ = 0.057, $BF_{01}$ = 11.67; RISC - $\tau_b$ = -0.078, $BF_{01}$ = 2.40; SPQ - $\tau_b$ = 0.006, $BF_{01}$ = 5.02).

## Modeling results
### Group level results
To quantitatively evaluate the relationships between underlying perceptual mechanisms and task performance we fitted a range of generative models. One class of models was Bayesian - it was

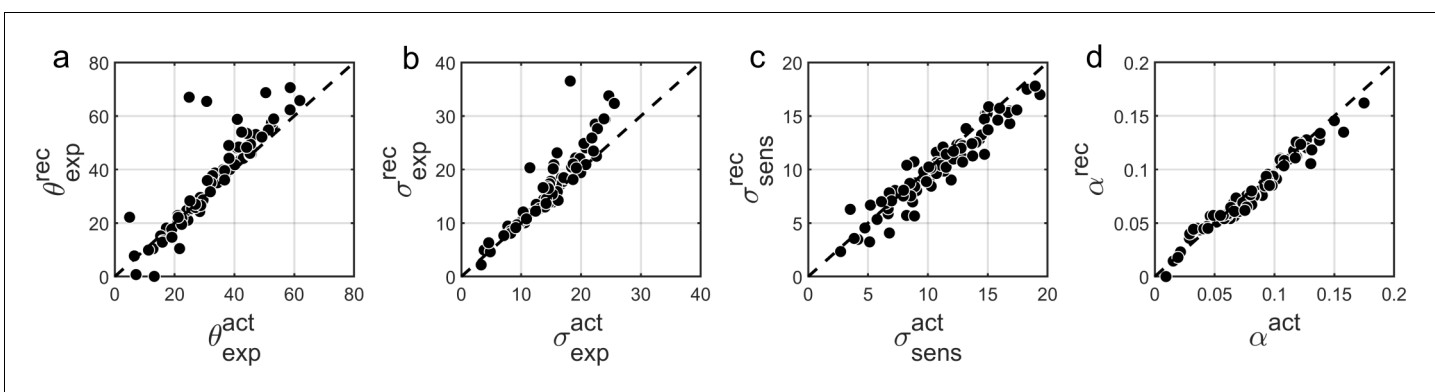

**Figure 8.** Comparison of actual (x-axis) vs. recovered (y-axis) parameters using the BAYES' model. (a) $\theta_{exp}$ - mean of the prior expectations (r = 0.90), (b) $\sigma_{exp}$ - uncertainty of the prior distribution (r = 0.92), (c) $\sigma_{sens}$ - uncertainty in the sensory likelihood (r = 0.95), (d) $\alpha$ - fraction of random estimations (r = 0.98). The dashed diagonal line is a reference line indicating perfect parameter recovery.
DOI: https://doi.org/10.7554/eLife.34115.012

based on the assumption that participants combine prior expectations with uncertain sensory information on a single trial basis (*Figure 5*).

To account for the possibility that the bimodal probability distribution of the stimuli, in addition to inducing prior expectations, has also affected the sensory likelihood, we constructed three variations of the Bayesian model: BAYES, where the sensory precision was constrained to be the same across all presented motion directions, 'BAYES_varmin, where the sensory precision was allowed to be different for the most frequently presented motion directions, but was the same across all other directions, and BAYES_var where sensory precision was allowed to be different across all motion directions. Another class of models was based on the assumption that task performance can be explained by response strategies that do not involve Bayesian inference. That is, on any given trial participants responded based on the prior expectations or sensory information alone. We considered four variations of response strategy models: ADD1, ADD2, ADD1_m, and ADD2_m (see Methods for details).

To compare the models, we computed BIC values for each individual for each model; we used individual BIC values as a summary statistic and compared the models using signed rank test in order to preserve individual variability, which corresponds to a random effects Bayesian model selection procedure. We found that the BAYES model had significantly smaller BIC values than the remaining models (see the p-values within *Figure 6a*).

To determine how the best fitting model compared to the actual data, we analyzed the estimation biases and variation in estimation responses as predicted by BAYES (*Figure 6b,c*). As in the experimental data analysis, we computed estimation distributions predicted by the model by assuming occasional random estimations (see *Equation 2*). Finally, using the BAYES model, we reconstructed the priors acquired by participants. While on the individual level there was a considerable variation in the shape of acquired priors (see *Appendix 1—figure 10*), on the group level, it approximated the statistics of the task (*Figure 6d*).

## Model parameters and autistic/schizotypal traits

Correlational analysis of BAYES model parameters showed that there was no correlation between AQ and the precision of the prior $\sigma_{exp}$ (*Figure 7b*; $r = 0.018$, p=0.962). That autistic traits had no effect on the precision of the prior was confirmed by Bayesian Kendall correlation, which provided positive evidence ($\tau_b = 0.001$, $BF_{01} = 6.99$).

Importantly, autistic traits were found to be strongly associated with less uncertainty in the sensory likelihood, $\sigma_{sens}$ (*Figure 7c*; $r = -0.185$, p=0.011), which also remained significant after Bonferroni correction (p=0.044). Finally, there was no correlation with the amount of random estimations (*Figure 7d*; $r = -0.135$, p=0.238). Motor precision, which was estimated from high contrast trials, separately from all other parameters (see Methods), was also correlated with autistic traits ($r = 0.245$, p=0.012). On the other hand, consistent with the absence of differences in the behavioral findings, schizotypal traits were not associated with any difference in the BAYES model parameter values (*Appendix 1—figure 9*), and in particular, were found to have no effect on prior precision (RISC: $\tau_b = -0.012$, $BF_{01} = 6.90$; SPQ: $\tau_b = 0.071$, $BF_{01} = 3.97$).

## Parameter recovery for BAYES

Finally, to further investigate that in our experimental paradigm the influence of stronger likelihoods can be distinguished from that of weaker priors (*Brock, 2012*; *Pellicano and Burr, 2012b*) we performed parameter recovery for the winning BAYES model. Parameter recovery involves generating synthetic data with different sets of parameters ('actual parameters') and then fitting the same model to estimate the parameters ('recovered parameters') that are most likely to have produced the data. If actual and recovered parameters are in a good agreement, it means that the effects of different parameters can be reliably distinguished. At the same time, parameter recovery is also affected by the parameter estimation methods and even more so by the amount of data used for model fitting. Therefore, parameter recovery provides an overall check for the reliability of modelling results and is recommended as an essential step in computational modelling approaches (*Palminteri et al., 2017*).

We found that overall BAYES model (and MLE parameter estimation using simplex optimization function) recovered parameters very well, which was reflected in Pearson's correlation between actual and recovered estimates being r > 0.9 for all model parameters (*Figure 8*).

## Discussion

In this study, we investigated whether autistic and schizotypal traits are associated with differences in the implicit Bayesian inference performed by the brain. Specifically, we wanted to know whether autistic and schizotypal traits are accompanied by (1) differences in how the priors are updated and/or in their precision and/or by (2) differences in the precision with which the sensory information (the likelihood) is represented. We used a visual motion estimation task (*Chalk et al., 2010*) that induces implicit prior expectations via more frequent exposure of two motion directions (±32°). We found that on the group level (N = 83) participants acquired prior expectations towards ±32° motion directions. This was indicated by shorter estimation reaction times and better detection at ±32°, as well as attractive biases towards ±32° and reduced estimation variability at ±32°. Moreover, when no stimulus was presented, participants sometimes still reported seeing the stimulus, mostly around ±32°. Performance was best explained by a simple Bayesian model, which provided a good fit to the data and captured the characteristic features of perceptual bias and variability. This model provided estimates of Bayesian priors and sensory likelihoods for each participant, which were then analyzed in relation to participants' schizotypal and autistic traits.

Schizotypal traits were found to have no measurable effect on perceptual biases in our task and, therefore, were not associated with any differences in the precision ascribed to priors and likelihoods. This finding challenges recent accounts of positive symptoms of schizophrenia that predict impaired updating of priors and an imbalance in precision ascribed to sensory information and prior expectations (*Fletcher and Frith, 2009*; *Corlett et al., 2009*; *Adams et al., 2013*). An immediate explanation might be that the influence of schizotypal traits in the healthy population is not strong enough to lead to behavioral differences, even if the dimensionality assumption holds. This would need to be addressed by further research investigating clinical populations. Another possibility is that the aberrant perception subconstruct of schizotypal traits, for which we did not acquire explicit measures, is more relevant for the hypothesized effects then the entire construct as a whole. For example, a recent study by *Powers et al. (2017)* found that overweighing of perceptual priors was specifically linked to hallucinatory propensity and not to the diagnostic status of psychosis itself. Furthermore, *Teufel et al. (2015)* also found that stronger influence of prior knowledge was primarily associated with hallucinatory propensity and not with delusional propensity. Another possible difference between *Teufel et al. (2015)* study and ours might be the level at which the priors operate. In *Teufel et al. (2015)* participants were presented with ambiguous two-tone versions of images before and after seeing the actual images in full color and had to report whether the presented two-tone image contains a face. The low-level prior for basic perceptual features (as induced in our task) might function at a hierarchically lower level than prior knowledge related to complex collection of features and semantic content (faces). The level at which prior expectations are induced has indeed been shown to matter. A series of studies by *Schmack et al. (2013, 2015, 2017)* using 3D rotating cylinders report weaker low-level (perceptually-induced - stabilizing) priors but stronger high-level (cognitively-induced) priors in both schizophrenia and schizotypal traits. It is difficult to compare and reconcile these findings with ours. One possibility is that the priors induced in our task lie in between their perceptual and cognitive levels. The taxonomy of priors in relation to their place in the computational hierarchy or to their complexity or specificity is still far from being established (*Seriès and Seitz, 2013*) and thus the potential relevance of such distinctions is still not known.

Autistic traits were associated with significant behavioral differences: weaker biases and lower variability of direction estimation on low contrast trials. Modeling revealed that this was because of increased sensory precision as well as higher motor precision, while there was no attenuation of acquired priors. Parameter recovery analysis confirmed that our methodology provides reliable parameter estimates and, in particular, allows disentangling variations in priors and likelihoods.

Autistic traits were also found to be associated with less false detections (hallucinations) on trials when no stimulus was presented, consistent with the idea that prior expectations had less influence in individuals with higher AQ. In an attempt to measure those individual differences, we fitted a more sophisticated Bayesian model that could account not only for the estimation performance but

also for the detection data (see Appendix 2). This model provided a good fit to both estimation and detection data, and preserved the correlation between ASD traits and the precision of the motion direction likelihood ($r = -0.202$, p=0.029). However, parameter recovery was not as good as for the BAYES model presented above (see *Appendix 2—figure 3*) and for this reason we focused on the simpler model in this paper.

Overall, our findings are in agreement with most of the recent Bayesian theories of ASD, namely, that autistic traits are associated with a relatively weaker influence of prior expectations. However, we find that this is due to enhanced sensory precision (*Lawson et al., 2014*; *Palmer et al., 2017*; *Brock, 2012*; *Van de Cruys et al., 2013*), rather than attenuated priors per se (*Pellicano and Burr, 2012a*). Other empirical studies inspired by the Bayesian accounts have reported either attenuated or intact priors, but most are subject to methodological limitations, either because they did not use computational modeling (*Skewes et al., 2015*; *Skewes and Gebauer, 2016*; *Croydon et al., 2017*) or because their model could not extract likelihoods and quantify their variations (*Powell et al., 2016*; *Lawson et al., 2017*).

The idea that sensory processing could be enhanced in autism has long been proposed outside the Bayesian framework. Autistic traits have been associated with enhanced orientation discrimination (*Dickinson et al., 2014*), but only for first-order (luminance-defined) stimulus (*Bertone et al., 2005*). This enhancement has been proposed to be a result of either enhanced lateral (*Bertone et al., 2005*), or a failure to attenuate sensory signals via top-down gain control (*Lawson et al., 2014*), both of which could be directly related to narrower likelihoods in the Bayesian framework (*Ma et al., 2006*). However, in motion perception, previous research did not find improved discrimination for first-order stimulus in autism, while for second-order (texture-defined) stimulus, the autistic group was found to underperform (*Bertone et al., 2003*). Our findings challenge these results and call for more research in this area.

In ASD as in schizotypy, prior integration might function differently at different levels of sensory processing. For example, *Pell et al. (2016)* reported intact direction-of-gaze priors for healthy individuals with high autistic traits and for highly functional individuals with a clinical diagnosis. The authors did not directly investigate differences in sensory precision, but the lack of behavioral differences suggests that there was none. Arguably, their paradigm involves more complex stimuli than used in our task, which are also strongly associated with semantic content (faces). It would not be surprising if increased sensory precision does not extend to such stimuli. In fact, autistic individuals are known to exhibit differential performance based on the complexity of the stimulus (*Bertone et al., 2005*), which also lies at the foundation of some theoretical accounts, such as the 'Weak Central Coherence' (*Happé and Frith, 2006*).

In our paradigm people acquire prior expectations very quickly, within 200 trials (see Appendix 1), which did not allow us to study individual differences in the rate at which the priors are acquired. Bayesian accounts predict differences in the dynamical updating of the priors, namely, that both autistic and schizotypal traits should be associated with increased learning rate - which is the ratio of likelihood and posterior precisions (*Palmer et al., 2017*). Our findings of increased sensory precision in autistic traits also suggest that their learning rate should be faster. However, this prediction might need to be more nuanced for volatile environments when there are multiple (hierarchical) levels of uncertainty that need to be updated simultaneously. A recent study by *Lawson et al. (2017)* found that when transitioning from stable to volatile environments, autistic adults showed larger change in the learning rate about volatility and smaller change in the learning rate about the environmental probabilities, while the average learning rates were found to not be different from those of controls.

Another aspect that our paradigm could not test is the specificity of the acquired priors (*Seriès and Seitz, 2013*). Some Bayesian accounts (*Van de Cruys et al., 2014*) predict that priors may be overly context-sensitive in autism. This is in line with the view that generalization is impaired in autism (*Plaisted, 2015*). Furthermore, such over-specificity is thought to be stronger with more repetitive stimuli (*Harris et al., 2015*). Future research could address this using statistical learning paradigms that incorporate increasingly distinct contexts or stimuli.

## Conclusion

We investigated statistical learning and Bayesian inference in a visual motion perception task along autistic and schizotypal traits. To our knowledge, this study is the first to investigate differences in Bayesian inference along both trait spectra in a single task. Furthermore, this study is the first visual

study to computationally disentangle and quantitatively assess the variations in individuals' likelihoods and priors. Surprisingly, schizotypal traits were found to have no effect on task performance and thus were not associated with any differences in the underlying statistical learning and Bayesian inference. For autistic traits, however, significant behavioral differences in prior integration were found, which were due to an increase in the precision of internal sensory representations in participants with higher AQ. Whether the current results extend to clinical populations will have to be examined in the future.

## Materials and methods

### Participants

91 (47 females, 44 males, age range: 18–69) naïve participants with no motor disabilities and with normal (or corrected to normal) vision were recruited from the general population. We advertised for participants using posters and the internet across University of Edinburgh locations and other sites across Edinburgh. All participants gave informed written consent and received monetary compensation for participation. The study was approved by the University of Edinburgh School of Informatics Ethics Panel.

### Questionnaires

ASD was assessed using 50-item version Autism Spectrum Quotient (AQ) (*Baron-Cohen et al., 2001*), which is commonly used for assessing milder variants of autistic-like traits within the general population. Schizotypal traits were assessed using The Rust Inventory of Schizotypal Cognitions (RISC) (*Rust, 1988*). RISC is specifically developed to measure schizotypal traits in the general population. In addition, a sub-group of 41 participants also completed Schizotypal Personality Questionnaire (SPQ) (*Raine, 1991*). Finally, all participants were also asked to complete the Warwick-Edinburgh Mental Well-being Scale (WEMWBS) (*Tennant et al., 2007*) in order to control for potential depression-induced differences in performance (*Austin et al., 2001*).

### Apparatus

The visual stimuli were generated using Matlab Psychophysics Toolbox (*Brainard, 1997*). Participants viewed the display in a dark room at a distance of 80–100 cm. The stimuli consisted of a cloud of dots with a density of 2 dots/deg$^2$ moving coherently (100%) at a speed of 9°/sec. Dots appeared within a circular annulus with minimum diameter of 2.2° and maximum diameter of 7°. The stimuli were displayed on a Dell P790 monitor running at 1024 × 768 at 100 Hz. The display luminance was calibrated using a Cambridge Research Systems Colorimeter (ColorCal MKII).

### The task

The task was developed previously in our laboratory (*Chalk et al., 2010*). Participants have to: (i) estimate the direction of coherently moving simple stimuli (dots) that are presented at low contrast levels (estimation task) and then (ii) indicate whether they have actually perceived the stimulus or not (detection task). Since *Chalk et al. (2010)* had shown that the effects of acquired priors become significant within the first 200 trials, instead of two experimental sessions of 850 trials each as in the original study, we used a single session of 567 trials (lasting around 40 min).

Each trial started by first displaying a fixation point (0.5°, 12.2 cd/m$^2$) for 400 ms, after which a field of moving dots appeared along with an orientation bar (length 1.1°, width 0.03°, luminance 4 cd/m$^2$, extending from the fixation point). Initial angle of the bar was randomized for each trial. Participants had to estimate the direction of motion by aligning the bar (using a computer mouse) to the direction the dots were moving in, and by clicking the mouse button to validate their estimate. The display cleared when either the participant had clicked the mouse or when 3000 ms had elapsed. On trials where no stimulus was presented, the bar still appeared for the estimation task to be completed.

After a 200 ms delay, the participants had to indicate whether they had actually detected the presence of dots in the estimation period (detection task). The display was divided into two parts by a vertical white line across the center of the screen, the left hand side area reading 'NO DOTS' and the right hand side area reading 'DOTS' (*Figure 2a*). The cursor appeared in the center of the

screen, and participants had to move it to the left or right and click to indicate their response. Immediate feedback for correct or incorrect detection responses was given by a cursor flashing green or red, respectively. The screen was cleared for 400 ms before the start of a new trial. Every 20 trials, participants were presented with feedback on their estimation performance in terms of average estimation error in degrees (e.g., 'In the last 20 trials, your average estimation error was 23°'). Every 170 trials (i.e. on three occasions) participants were given a chance to 'have a short break to rest their eyes', in order to prevent fatigue. Participants clicked when they were ready to continue.

## Design

The stimuli were presented at four different levels of contrast: 0 contrast (no-stimulus trials), two low levels contrasts and high contrast, randomly mixed across trials. There were 167 trials with no stimulus. The two low levels of contrast were determined using 4/1 and 2/1 staircases on detection performance (*García-Pérez, 1998*). There were 243 trials following the 4/1 staircase and 90 trials following the 2/1 staircase. The remaining 67 trials were at high contrast, which was set to 3.51 cd/m² above the background luminance.

For the two low contrast levels, there was a predetermined number of possible directions: 0°, ±16°, ±32°, ±48°, and ±64° with respect to a reference direction. The reference direction was randomized for each participant. For the 2/1 staircased contrasts, each predetermined motion direction was presented equally frequently. Unbeknownst to participants, stimuli at high and 4/1 staircase contrasts were presented more frequently at −32° and +32° motion directions, resulting in a bimodal probability distribution (*Figure 1b*). For the 4/1 staircase contrast level, the dots were moving at ±32° in 173 (~70%) trials and in all the other predetermined motion directions in the remaining 70 (~30%) trials equally frequently. At the highest contrast level, 34 (~50%) trials had the dots moving at ±32° and the remaining 33 (~50%) trials were at random directions (i.e. not just the predetermined directions).

## Data analysis

Responses on high contrast trials were used as a performance benchmark to ensure that participants were performing the task adequately. The predefined inclusion criteria were: (1) at least 80% detection and (2) less than 30° root mean squared error of estimations. 8 out of 91 participants failed to satisfy at least one of the criteria and were excluded from further analysis (*Appendix 1—figure 1*).

Data analysis on the estimation of motion directions was performed on 4/1 and 2/1 staircased contrast levels only and only on trials where participants both validated their choice with a click within 3000 ms in the estimation part and clicked 'DOTS' in the detection part. The first 170 trials of each session were excluded from the analysis, as this was the upper limit for the convergence of the staircases to stable contrast levels (*Appendix 1—figure 2*).

After removing these trials, the luminance levels achieved by the 2/1 and 4/1 staircases were found to be considerably overlapping (*Appendix 1—figure 2*). Therefore, the data for both of these contrast levels was combined for all further analysis.

To account for random estimations (either accidental or intentional) that participants made on some trials, we fitted each participant's estimation responses to the probability distribution:

$$(1 - \alpha) \cdot V(\theta | \mu, \kappa) + \alpha, \tag{2}$$

Where $\alpha$ is the proportion of trials in which participant makes random estimates, and $V(\theta|\mu,\kappa)$ is the probability density function for the estimated angle $\theta$ for von Mises (circular normal) distribution with the mean μ and precision $\kappa$. The parameters μ and $\kappa$ of the von Mises distribution were determined by maximizing the likelihood of the distribution in *Equation (2)* for each presented angle.

To analyze the distribution of estimations in no-stimulus trials, we constructed histograms of 16° size bins. These histograms were converted into probability distributions by normalizing over all motion directions. We analyzed the estimation distribution when participants reported seeing dots (clicked 'DOTS') within no-stimulus trials. We interpreted these false alarms as a simple form of perceptual hallucination.

## Modelling

### Bayesian models

Bayesian models assume that participants combined a learned prior of the stimulus directions with their sensory evidence in a probabilistic manner. We first assume that participants make noisy sensory observations of the actual stimulus motion direction ($\theta_{act}$), with a probability

$$\mathrm{p}_{sens}(\theta_{sens}|\theta_{act}) = V(\theta_t, \kappa_{sens}). \tag{3}$$

where $\theta_t$ itself varies from trial to trial around θact according to p($\theta_t$|$\theta_{act}$)=V($\theta_{act}$, $\kappa_{sens}$).

While participants cannot access the 'true' prior, p($\theta$), directly, we hypothesized that they learned an approximation of this distribution, denoted $p_{exp}$($\theta$). This distribution was parameterized as the sum of two von Mises distributions, centered on motion directions $\theta_{exp}$ and -$\theta_{exp}$, and each with precision $\kappa_{exp}$:

$$\mathrm{p}_{exp}(\theta) = 0.5[V(-\theta_{exp}, \kappa_{exp}) + V(\theta_{exp}, \kappa_{exp})] \tag{4}$$

Combining these via Bayes' rule gives a posterior probability that the stimulus is moving in a direction $\theta$:

$$\mathrm{p}_{post}(\theta|\theta_{sens}) \propto \mathrm{p}_{exp}(\theta) \cdot \mathrm{p}_{sens}(\theta_{sens}|\theta) \tag{5}$$

The perceived direction, $\theta_{perc}$, was taken to be the mean of the posterior distribution (almost identical results would be obtained by using the maximum instead). Finally, we accounted for motor precision and a possibility of random estimates on some trials via:

$$p(\theta_{est}|\theta_{perc}) = (1-\alpha) \cdot V(\theta_{perc}, \kappa_m) + \alpha, \tag{6}$$

where $\alpha$ is the proportion of trials in which participants make random estimates and $\kappa_m$ is the motor precision.

Increased exposure to some motion directions might not only give rise to prior expectations, but also induce learning in the sensory likelihood function itself (*Stocker and Simoncelli, 2006*; *Sato and Kording, 2014*). Therefore, we fitted two more model variants: 'BAYES_var' where $\kappa_{sens}$ varied with the stimulus direction (i.e. it took five different values for each of the angles: 0°, ±16°, ±32°, ±48°, ±64°) and 'BAYES_varmin' where $\kappa_{sens}$ was allowed to be different for ±32° but was the same for all other directions.

### Response strategy models

We wanted to test whether task behavior might be better explained by simple behavioral strategies. This class of models assumed that on trials when participants were unsure about the presented motion direction, they made an estimation based solely on prior expectations, while on the remaining fraction of trials they made unbiased estimates based solely on sensory inputs. The first model, 'ADD1', assumed that estimations derived from prior expectations were simply sampled from a learnt expected distribution, $p_{exp}$($\theta$) (see *Chalk et al., 2010* and Appendix 2). The second model, 'ADD2', was just as 'ADD1' except when participants were unsure about the stimulus motion direction, instead of sampling from the complete learned probability distribution ranging from −180° to +180°, they effectively truncated this distribution on a trial by trial basis and sampled from only one part of it, negative (−180° to 0°) or positive (0° to +180°), depending on which side of the distribution the actual stimulus occurred (see *Chalk et al., 2010*) and Appendix 2). We also considered slight variations of the 'ADD1' and 'ADD2' models, denoted 'ADD1_m' and 'ADD2_m' respectively. These were identical to 'ADD1' and 'ADD2' except from setting 1/$\kappa_{exp}$ to zero; that is, on trials when perceptual estimates were derived only from expectations, they were equal to the mode of the learnt distribution (i.e. no uncertainty).

### Parameter estimation

We used performance in high contrast trials to estimate motor precision, $\kappa_m$, for each individual. We assumed that, for those trials, sensory uncertainty was close to zero. Motor precision was then determined by fitting estimation responses to the distribution in *Equation (2)* by replacing μ with the

actual motion direction, $\theta_{act}$. The estimated motor precision was used in all subsequent model fitting as a fixed parameter. The rest of the free parameters were estimated by fitting the response data at the two low (staircased) contrast levels. For each model with a set of free parameters $M$, we computed the probability distribution $p(\theta_{est}|\theta_{act}; M)$ of making an estimate $\theta_{est}$ given the actual stimulus direction $\theta_{act}$. For the response strategy models, by definition, the $p(\theta_{est}|\theta_{act}; M)$ corresponds to average behavior in the task.

The parameters were estimated by maximizing the fit of the log likelihood function for the experimental data for each participant individually. The maximum likelihood was found using a simplex algorithm, using *fminsearchbnd* Matlab function. To avoid convergence at a local maximum we constructed a grid of initial $\kappa_{exp}$ and $\kappa_{sens}$ parameter values covering the range found in previous studies. We selected the resulting set of parameters that corresponded to the largest log-likelihood.

## Model comparison

To compare the model fits we used Bayesian Information Criterion (BIC), which approximates the log of model evidence (*Burnham and Anderson, 2004*):

$$-2 \cdot log(P(D|M)) \approx BIC = -2 \cdot log(P(D|M,\hat{\Theta})) + k \cdot log(n), \tag{7}$$

where M is model, D is observed data and $P(D|M, \hat{\Theta})$ is the likelihood of generating the experimental data given the most likely set of parameters, $\hat{\Theta}$; $k$ is the number of model parameters and $n$ is the number of data points (or equivalently, the number of trials). BIC evaluates the model by how it fits the data by also penalizing for model complexity (number of parameters); lower BIC score indicates a better model.

## Parameter recovery

To determine whether the BAYES model can distinguish the effects of strong likelihoods from those of weak priors (*Brock, 2012*; *Pellicano and Burr, 2012b*) and to evaluate the robustness of our methods, we performed parameter recovery. First, we generated 80 sets of parameters (i.e. 80 synthetic individuals) by randomly sampling each parameter from a Gaussian distribution centered on the mean value of each parameter found in our sample (40° for $\theta_{exp}$, 15° for $\sigma_{exp}$, 10° for $\sigma_{sens}$, 0.06 for $\alpha$ and 10° for $\sigma_{motor}$). Second, for each set of parameters, we simulated data for 200 trials with the Bayesian model by randomly sampling from the estimation probability distribution. We used 200 simulated trials only, to match the empirical data (200 corresponds to the amount of experimental trials used for fitting, after excluding high contrast and zero contrast trials; Simulating more trials would result in a better parameter recovery but the results would no longer be informative about the reliability of parameters estimated from empirical data). Finally, we fitted the BAYES model to the simulated data. To evaluate the goodness of recovered parameters, we computed Pearson's correlation between the actual parameters and the recovered parameters.

## Statistical tests

Due to the presence of outliers in many of the measures, we used robust regression techniques for measuring the presence and strength of the effects in our data. This was done using *robustfit* function in Matlab, which downweighs the influence of outliers in proportion to their distance from the regression line, which is computed via iteratively reweighted least squares (IRLS) (*Holland and Welsch, 1977*). For the loss function we used Huber function (*Huber, 1964*) with a tuning constant of 1.345, which corresponds to 95% estimator efficiency as compared to ordinary least squares.

Furthermore, we applied Bonferroni correction for multiple testing based on the number of independent hypotheses that we tested; that is, whether two personality traits, ASD and schizotypy, were associated with the two variables of interest, acquired priors and sensory likelihoods, - this resulted in four different hypotheses. Note that while the number of null hypothesis significance tests that we performed exceeds this number, the tests within each set concerning the same hypothesis were not independent (each test was based on derivative and/or correlated values to those in the other tests within the same set), and thus would not have met the independence assumption on which Bonferroni correction is based.

Finally, due to the limitations of frequentist statistics for accepting the null hypothesis, we performed Bayesian correlation analysis and computed Bayesian Factors (*Kass and Raftery, 1995*) for

the null hypothesis ($BF_{01}$). This was done using JASP (*Team, 2017*) (Version 0.8.6). Due to the presence of outliers, this analysis was carried out using the non-parametric Kendall's Tau-b correlation coefficient.

## Source code and data

The source data of the main figures is provided. These include, *Figure 3—source data 1*, *Figure 4—source data 1* and *Figure 7—source data 1*. *Source code 1* contains all the source code necessary to reproduce the figures. More detailed information about the source code is in SourceCode_Readme.txt, while SourceData_Readme.txt contains more details about the source data files.

## Acknowledgements

We thank Gizem Aras for assisting in data collection, and Katie Richards for assisting with participants' recruitment. This work was done in part while PS was visiting the Simons Institute for the Theory of Computing.

## Additional information

### Funding

| Funder | Grant reference number | Author |
| --- | --- | --- |
| Brain and Behavior Research Foundation | NARSAD Young investigator grant 19271 | Peggy Seriès |

The funders had no role in study design, data collection and interpretation, or the decision to submit the work for publication.

### Author contributions

Povilas Karvelis, Formal analysis, Investigation, Visualization, Writing—original draft; Aaron R Seitz, Software, Funding acquisition, Methodology, Writing—review and editing; Stephen M Lawrie, Methodology, Writing—review and editing; Peggy Seriès, Conceptualization, Supervision, Project administration, Writing—review and editing

### Author ORCIDs

Stephen M Lawrie http://orcid.org/0000-0002-2444-5675
Peggy Seriès https://orcid.org/0000-0002-8580-7975

### Ethics

Human subjects: All participants gave informed written consent and received monetary compensation for participation. The study was approved by the University of Edinburgh School of Informatics Ethics Panel.

### Decision letter and Author response

Decision letter https://doi.org/10.7554/eLife.34115.031
Author response https://doi.org/10.7554/eLife.34115.032

## Additional files

### Supplementary files

• Source code 1. Matlab scripts for data analysis and reproduction of the figures presented in the article.
DOI: https://doi.org/10.7554/eLife.34115.013

• Transparent reporting form
DOI: https://doi.org/10.7554/eLife.34115.014

## Data availability

All data generated or analysed during this study are included in the manuscript and supporting files. Source code has been included, which can be used to reproduce the results figures.

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

# Appendix 1

DOI: https://doi.org/10.7554/eLife.34115.015

## Exclusion criteria

In order to ensure that participants performed adequately in the psychophysical task, we used predetermined performance criteria for inclusion into the study. Firstly, participants were required to detect the motion stimuli on more than 80% of trials with the high contrast motion stimuli and also make active estimates of the motion directions by clicking the mouse. Secondly, their average estimation performance on the high contrast stimuli had to be within 30° of the correct angle. 8 out of 91 participants failed to satisfy at least one of the criteria: two participants did not satisfy the first criteria, four did not satisfy the second criteria and two did not satisfy both of the criteria (*Appendix 1—figure 1*). These participants were excluded from further analysis.

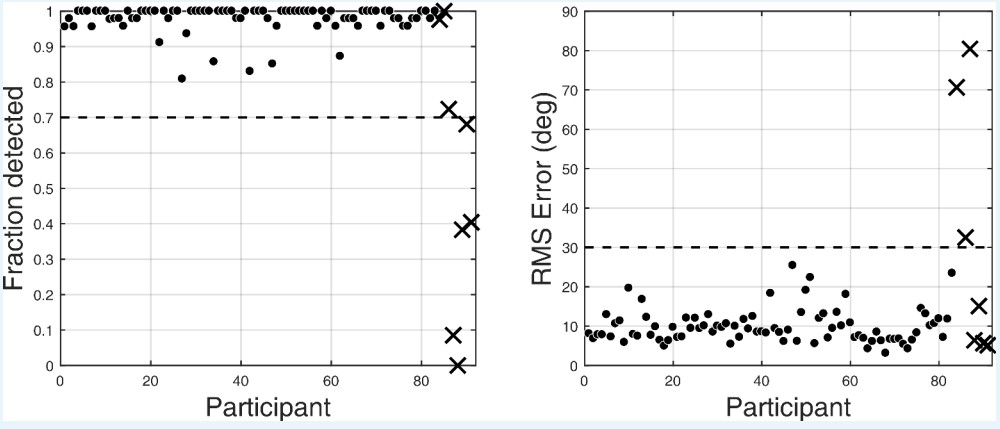

**Appendix 1—figure 1.** Task performance at the highest contrast level and exclusion Criteria. Left panel: fraction of detected high contrast trials - quantified as the fraction of trials in which participants both validated their choice with a click within 3000 ms in the estimation part and reported seeing dots (clicked 'DOTS') in the detection part. Right panel: root mean square error of estimations on high contrast trials. The dashed lines represent minimum performance criteria (more than 80% detection and less than 30° RMS error of estimations). Excluded participants are denoted by cross markers.

DOI: https://doi.org/10.7554/eLife.34115.016

## Staircased stimulus contrast levels

*Appendix 1—figure 2* describes the average convergence of the contrast staircases. Two groups comprising our sample performed the task at different background contrast levels. For a subgroup of 50 participants (left panel), the background luminance was set to 1.16 cd/m$^2$ for the other sub-group of 41 (right panel) it was set to 5.18 cd/m$^2$. For both groups, contrast staircases converged after 170 trials for both intermediate contrast levels, denoted with the vertical dashed line. In both groups, 2/1 and 4/1 staircased contrasts were considerably overlapping: on average 2/1 being 0.20 ± 0.04 cd/m$^2$ and 4/1 being 0.22 ± 0.04 cd/m$^2$ above the 1.16 cd/m$^2$ background luminance; and on average 2/1 being 0.42 ± 0.05 cd/m$^2$ and 4/1 being 0.46 ± 0.05 cd/m$^2$ above the 5.18 cd/m$^2$ background luminance. Thus, the two intermediate contrasts were combined for all further data analysis.

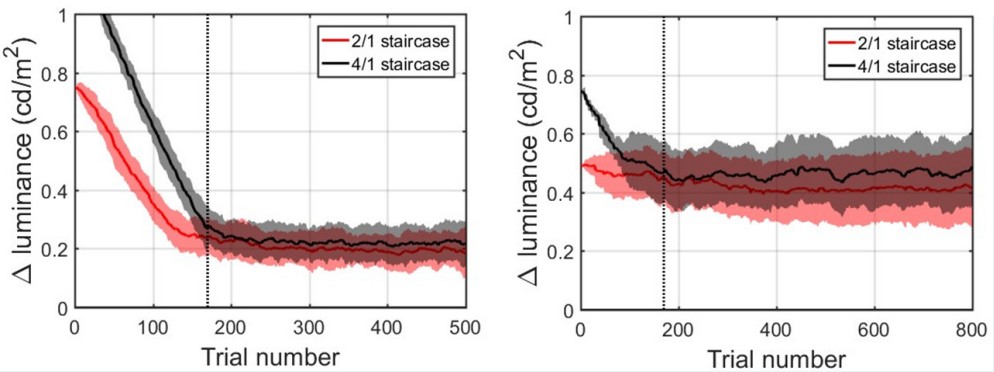

**Appendix 1—figure 2.** Population averaged stimulus contrast relative to the background contrast for the 2/1 (red) and 4/1 (black) staircased contrast levels. Standard deviation is denoted by shaded areas with corresponding colors. The vertical dashed line marks 170 trials. Left panel: 44 participants (remaining after exclusion) that performed the task with the background luminance set to 1.16 cd/m$^2$. Right panel: 39 participants (remaining after exclusion) that performed the task with the background luminance set to 5.18 cd/m$^2$.

DOI: https://doi.org/10.7554/eLife.34115.017

## Combining the different background luminance levels

To compare the two sub-groups that performed the task at different background luminance levels, we performed Wilcoxon two-tailed rank sum test for all of the behavioral measures and none of them indicated any differences: mean absolute estimation bias (z = 0.652; ranksum = 1920; p=0.514), mean variance of estimations (z = −0.406; ranksum = 1803; p=0.685), total number of hallucinations (z = 0.128; ranksum = 1862; p=0.898) number of hallucinations within 8° of ±32° (z = 0.870; ranksum = 1943; p=0.384), mean estimation reaction time (z = 0.479; ranksum = 1901; p=0.632). The two groups were therefore combined.

## Temporal emergence of the impact of expectations

We investigated how many trials it took for the acquired prior effects to impact behavior. First, we looked at estimation reaction times (RT) and compared mean RT of each individual at ±32° with mean RT at all other directions; we compared cumulative moving averages at every 30 trials (*Appendix 1—figure 3*). We found that it took less than 90 trials for RT at ±32° to become significantly shorter than average RT at all other directions (*Appendix 1—figure 3* and p-values within).

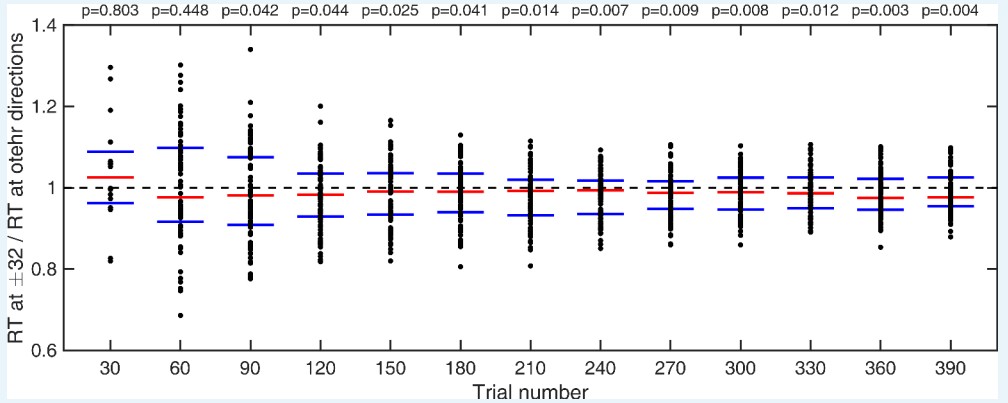

**Appendix 1—figure 3.** Cumulative moving average of ratio of estimation reaction times at ±32° vs average reaction times at all other directions. Red bars indicate median values and blue bars indicate 25th and 75th percentiles. p-values indicate whether RTs at ±32° are significantly shorter than average RTs over all other directions (one-tailed Wilcoxon signed rank test).

DOI: https://doi.org/10.7554/eLife.34115.018

Similarly, we looked at average detection performance and compared the fraction of trials in which stimulus was detected at ±32° with the mean fraction detected over all other presented directions; again, we compared cumulative moving averages at every 30 trials (*Appendix 1—figure 4*). We found that it took less than 90 trials for detection at ±32° to become significantly better than average detection over all other presented directions (*Appendix 1—figure 4* and p-values within).

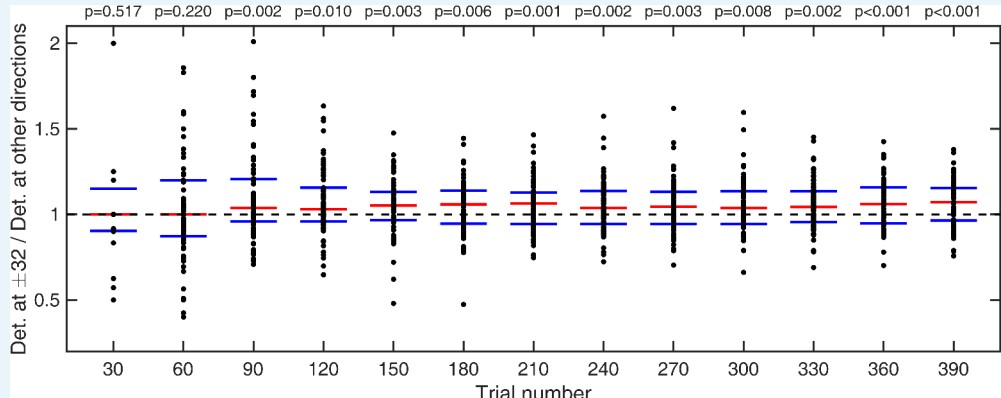

**Appendix 1—figure 4.** Cumulative moving average of ratio of fraction of detected stimuli at ±32° vs average fraction detected at all other directions. Red bars indicate median values and blue bars indicate 25th and 75th percentiles. p-values indicate whether fraction detected at ±32° are significantly larger than average fraction detected over all other directions (one-tailed Wilcoxon signed rank test).

DOI: https://doi.org/10.7554/eLife.34115.019

Lastly, for trials where no stimulus was presented, we looked at how long it took participants to start hallucinating predominantly around ±32° as opposed to all other possible directions. This was quantified as a probability ratio $p_{rel}$:

$$p_{rel} = p(\theta_{est} = \pm 32(\pm 8)^\circ) \cdot N_{bins}, \tag{1}$$

where $N_{bins}$ is the number of bins (45), each of size 16°. This probability ratio would be equal to one if participants were equally likely to estimate within 8° of ±32° as they were to estimate within other bins. Again, we computed cumulative moving mean at every 30 trials (*Appendix 1—figure 5*). For participants who did not report seeing dots at any direction within a given number of trials (i.e. zero total hallucinations) this probability ratio was undefined, therefore, those individuals were omitted from significance test at that point. We found that it took less than 210 trials for $p_{rel}$ to become significantly larger than 1 (*Appendix 1—figure 5* and p-values within).

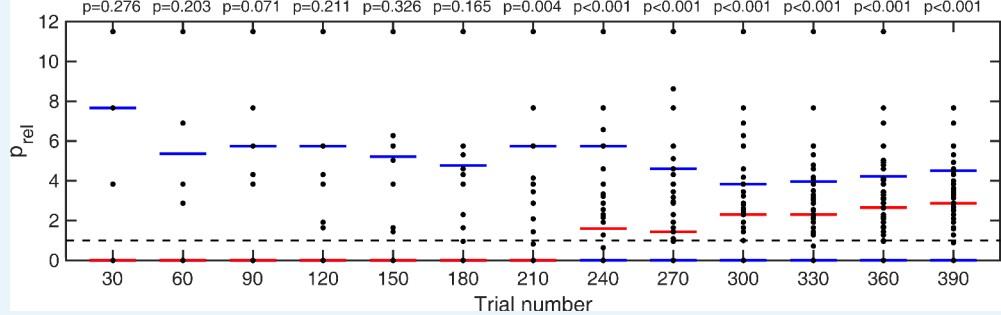

**Appendix 1—figure 5.** Cumulative moving average of ratio of fraction of detected stimuli at ±32° vs average fraction detected at all other directions. Red bars indicate median values and blue bars indicate 25th and 75th percentiles. p-values indicate whether fraction detected

at ±32° are significantly larger than average fraction detected over all other directions (one-tailed Wilcoxon signed rank test).

DOI: https://doi.org/10.7554/eLife.34115.020

## Schizotypy traits and task performance

*Appendix 1—figure 6* and *Appendix 1—figure 7* show task performance by groups which were formed by splitting the sample on the median RISC and SPQ scores respectively. *Appendix 1—figure 8* shows the correlations between RISC and SPQ scores and the corresponding performance measures. There were no significant correlations with any of the measures.

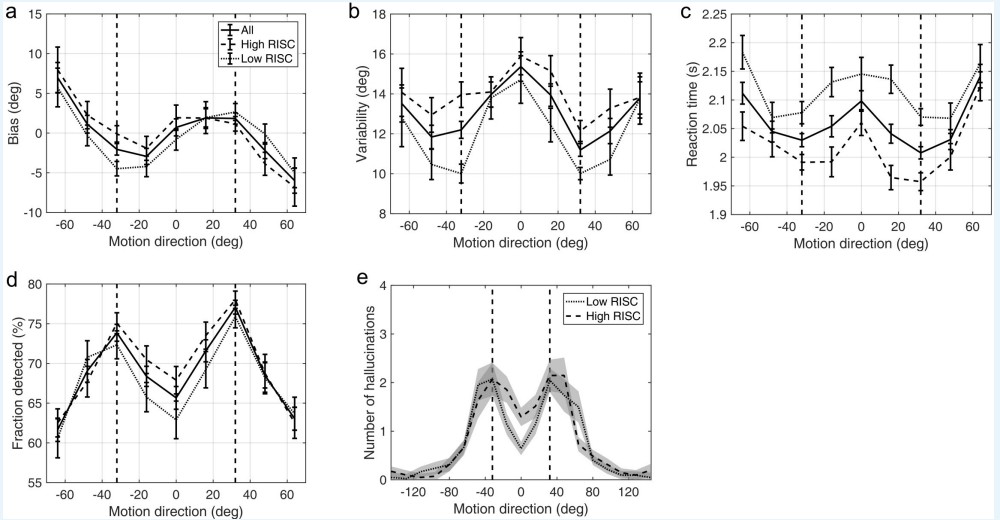

**Appendix 1—figure 6.** Average group performance on low-contrast trials (**a–d**) and on trials with no stimulus (**e**) by groups split by median RISC score. (**a**) Mean estimation bias, (**b**) standard deviation of estimations, (**c**) estimation reaction time and (**d**) fraction of trials in which the stimulus was detected. (**e**) Distribution of hallucinations. The vertical dashed lines correspond to the two most frequently presented motion directions (±32°). Error bars and shaded areas represent within-subject standard error.

DOI: https://doi.org/10.7554/eLife.34115.021

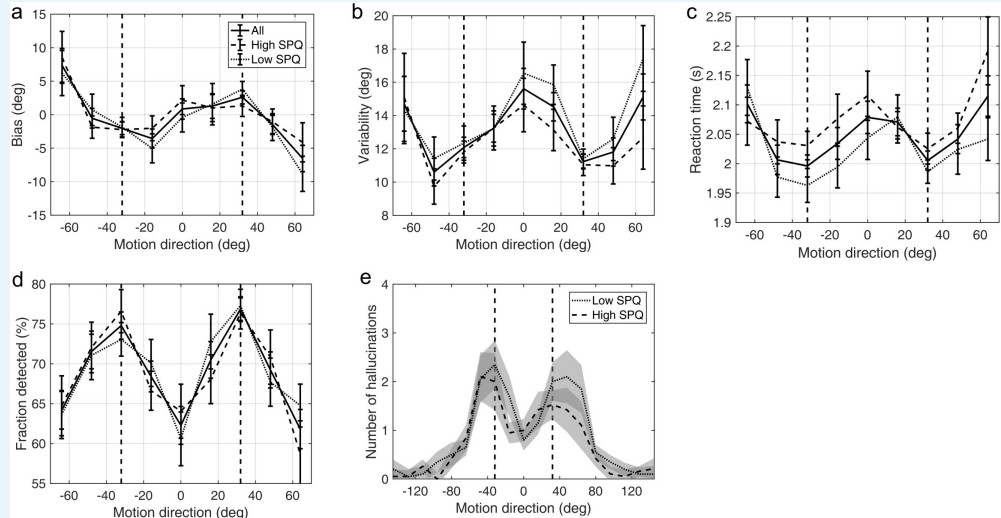

**Appendix 1—figure 7.** Average group performance on low-contrast trials (**a–d**) and on trials with no stimulus (**e**) by groups split by median SPQ score. (**a**) Mean estimation bias, (**b**) standard deviation of estimations, (**c**) estimation reaction time and (**d**) fraction of trials in which the

stimulus was detected. (**e**) Distribution of hallucinations. The vertical dashed lines correspond
to the two most frequently presented motion directions (±32°). Error bars and shaded areas
represent within-subject standard error.

DOI: https://doi.org/10.7554/eLife.34115.022

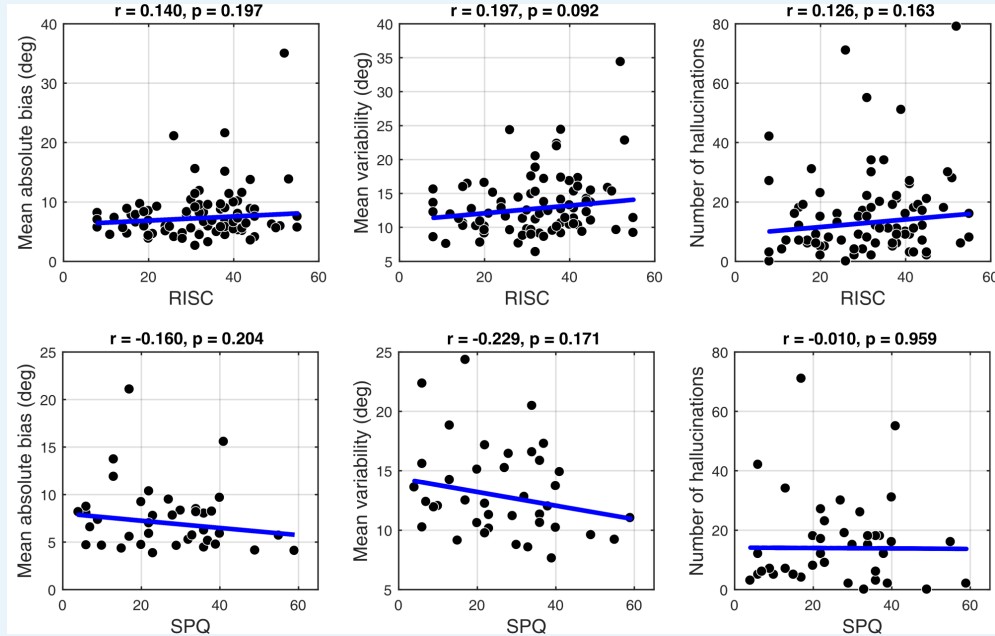

**Appendix 1—figure 8.** Correlations between personality traits, RISC (top row) and SPQ (bottom row) and task performance. There were no significant correlations with any of the measures: mean absolute bias (left column), mean estimation variability (middle column) and total number of hallucinations (right column). Robust correlation coefficients and p-values are indicated above each plot. The blue lines denote robust regression.

DOI: https://doi.org/10.7554/eLife.34115.023

## Schizotypy traits and model parameters

*Appendix 1—figure 9* shows the robust correlation analysis results between the BAYES model parameter estimates and schizotypy scores. There was no significant correlation with any of the parameters. Further Bayesian correlation analysis provided positive evidence that schizotypy traits had no effect on prior precision (RISC: $\tau_b$ = -0.012, $BF_{01}$ = 6.90; SPQ: $\tau_b$ = 0.071, $BF_{01}$ = 3.97).

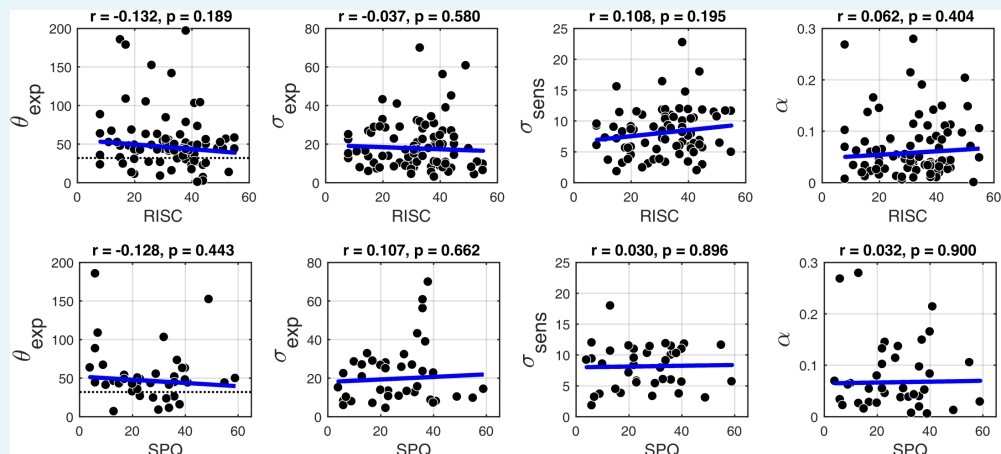

**Appendix 1—figure 9.** Correlations with the BAYES model parameter values and schizotypy traits (as measured by both RISC and SPQ). First column: $\theta_{exp}$ - mean of the prior expectations,

second column: $\sigma_{exp}$ - uncertainty of the prior distribution, third column: $\sigma_{sens}$ - uncertainty in the sensory likelihood and fourth column: $\alpha$ - fraction of random estimations. Robust correlation coefficients and p-values are indicated above each plot. The blue lines denote robust regression.

DOI: https://doi.org/10.7554/eLife.34115.024

## Individual priors recovered via BAYES model

*Appendix 1—figure 10* shows a representative sample of the priors we extracted for a number of individuals, using the 'BAYES' model.

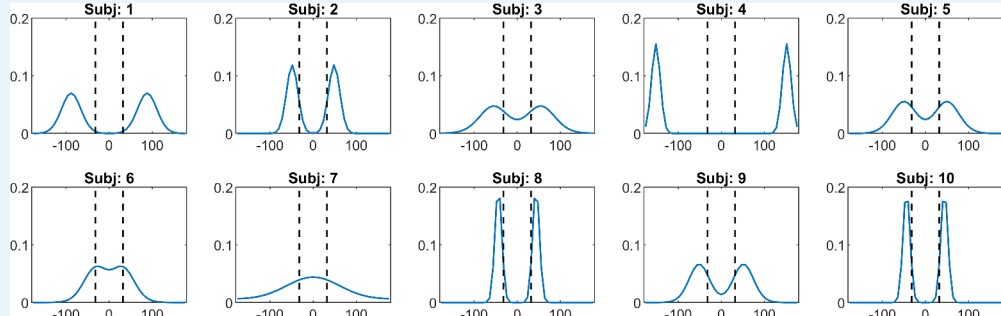

**Appendix 1—figure 10.** A representative sample of prior expectations for each individual as reconstructed via 'BAYES' model. The dashed lines correspond to the two most frequently presented motion directions (±32°).

DOI: https://doi.org/10.7554/eLife.34115.025

## Appendix 2

DOI: https://doi.org/10.7554/eLife.34115.026

### Response bias models

We wanted to account for the possibility that the task behavior might be better explained by simple behavioral strategies. This class of models assumed that on trials when participants were unsure about the presented motion direction they made an estimation based solely on prior expectations, while on the remaining fraction of trials they made unbiased estimates based solely on sensory input.

### ADD1

The first model ('ADD1') assumed that when participants were unsure about which motion direction they had perceived, they made an estimate that was close to one of the two most frequently presented motion directions. In this model, on each trial, participants make a sensory observation of the stimulus motion direction, $\theta_{sens}$. We parameterize the probability of observing the stimulus to be moving in a direction $\theta_{sens}$ by a von Mises (circular normal) distribution centered on the actual stimulus direction and with width determined by $1/\kappa_{sens}$:

$$p_{sens}(\theta_{sens}|\theta_{act}) = V(\theta_{act}, \kappa_{sens}) \tag{2}$$

On most trials, we assume that participants make a perceptual estimate of the stimulus motion direction ($\theta_{perc}$) that is based entirely on their sensory observation so that $\theta_{perc} = \theta_{sens}$. However, on a certain proportion of trials, when participants are uncertain about whether a stimulus was present or not, they resort to their expectations by making a perceptual estimate that is sampled from a learned distribution, $p_{exp}(\theta)$. For simplicity, we parameterize this distribution as the sum of two circular normal distributions, each with width determined by $1/\kappa_{exp}$, and centered on motion directions $-\theta_{exp}$ and $\theta_{exp}$, respectively. Finally, we accommodate for the fact that there will be a certain amount of noise associated with moving the estimation bar to indicate which direction the stimulus is moving in as well as allowing for a fraction of trials $\alpha$, where participants make estimates that are completely random. Thus, the estimation response $\theta_{est}$ is related to the perceptual estimate $\theta_{perc}$ via the equation:

$$p(\theta_{est}|\theta_{perc}) = (1-\alpha)^{\star}V(\theta_{perc}, \kappa_m) + \alpha. \tag{3}$$

Bringing all this together, the distribution of estimation responses for a single participant is given by:

$$p(\theta_{est}|\theta_{act}) = (1-\alpha)[(1-a(\theta))p_{sens}(\theta_{sens} = \theta_{est}|\theta_{act}) + a(\theta)p_{exp}(\theta)]^{\star}V(0, \kappa_m) + \alpha. \tag{4}$$

where the asterisk denotes a convolution and a($\theta$) determines the proportion of trials that participants sampled from the expected distribution, $p_{exp}(\theta)$. The resulting 'ADD1' model has nine free parameters $\theta_{exp}$, $\kappa_{exp}$, a($\theta$) (which can take a different value for each of the five angles: 0°,±16°,±32°,±48°,±64°), $\kappa_{sens}$ and $\alpha$.

### ADD2

The second model, 'ADD2', was just as 'ADD1' except that it had slightly more complex strategy for trials when participants were unsure about the stimulus motion direction: instead of sampling from the complete learned probability distribution ranging from $-180°$ to $+180°$ (*Equation (11)*), they effectively truncated this distribution on a trial by trial basis and sampled from only one part of it, negative ($-180°$ to $0°$) or positive ($0°$ to $+180°$), depending on which side of the distribution the actual stimulus occurred. Incorporating this into the distribution of estimation responses gives:

$$p(\theta_{est}|\theta_{act}) = (1-\alpha)[(1-a(\theta)-b(\theta))p_{sens}(\theta_{sens} = \theta_{est}|\theta_{act}) \\ +a(\theta)p_{expN}(\theta)+b(\theta)p_{expP}(\theta)]*V(0,\kappa_m)+\alpha. \tag{5}$$

where asterisk (*) denotes convolution; $a(\theta)$ and $b(\theta)$ determine the proportion of trials in which participants sample from either anticlockwise or clockwise distributions $p_{expN}(\theta)$ and $p_{expP}(\theta)$, respectively.

In addition, we also considered slight variations of the 'ADD1' and 'ADD2' models, denoted 'ADD1_m' and 'ADD2_m' respectively. These were identical to 'ADD1' and 'ADD2' except from setting $1/\kappa_{exp}$ to zero; that is, on trials when perceptual estimates were derived only from expectations, they were equal to the mode of the learnt distribution (i.e. no uncertainty).

## Non-symmetric prior models

The stimulus distribution is multimodal and symmetric. Learning such a distribution might be inherently difficult. We reasoned that some individual differences might lie in asymmetries of the acquired priors. Therefore, we explored an alternative parameterization of the acquired priors which allowed them to be asymmetrical. We allowed the two modes in the prior to have different position with respect to $0°$ and to have different amount of probability associated with each mode. This resulted in:

$$p_{exp}(\theta) = (1-\pi) \cdot V(\theta_p, \kappa_{exp}) + \pi \cdot V(\theta_n, \kappa_{exp}) \tag{6}$$

where $\pi$ ($\in [0\ 1]$) is a mixing parameter. Using this parameterization we fitted 'BAYES' model as described in the main text (thus, we denoted this alternative model as 'BAYES_$\pi$'). The alternative parameterization did not result in a better BIC as compared to 'BAYES' model (p=0.378, signed rank test). In addition, we performed parameter recovery to determine how robust 'BAYES_$\pi$' is and found that recovering the mixing parameter $\pi$ was not very reliable (r = 0.4), although other parameters retained most of their previous reliability (*Appendix 2—figure 1*). We thus focused on the simpler model in the current study.

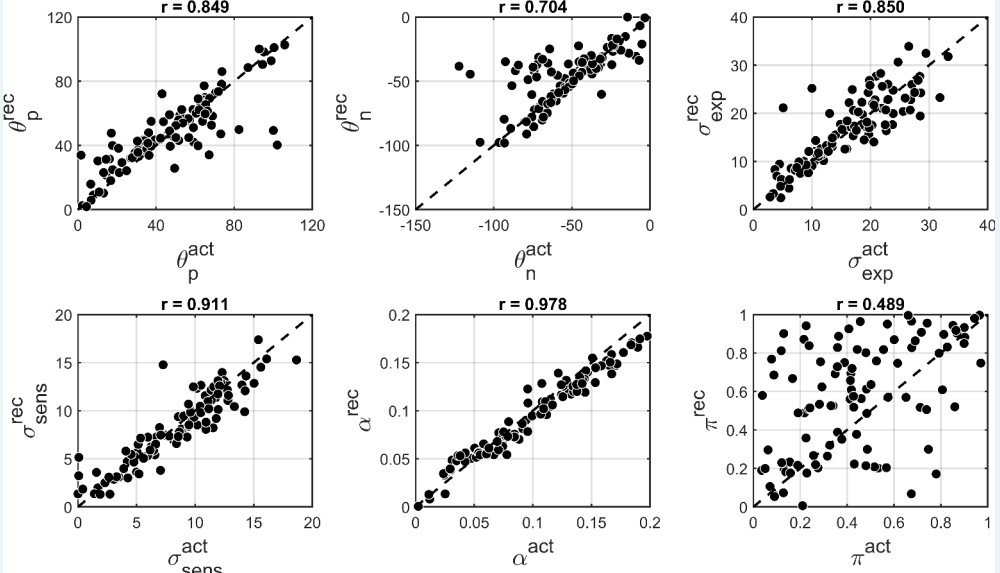

**Appendix 2—figure 1.** Comparison of actual and recovered parameters via 'BAYES_$\pi$' model. $\theta_p$ and $\theta_n$ - positive and negative modes of the bimodal distribution of prior expectations, $\sigma_{exp}$ - uncertainty of the prior distribution, $\sigma_{sens}$ uncertainty in the sensory likelihood, $\alpha$ - fraction of random estimations, $\pi$ - mixing parameter responsible for the degree of bimodality. Actual parameters are scattered along x-axis and recovered parameters are scattered along y-axis.

The dashed diagonal line is a reference line indicating perfect parameter recovery. Pearson's correlation coefficients are indicated above each plot.

DOI: https://doi.org/10.7554/eLife.34115.027

## Full models (estimation + detection)

We have built a Bayesian model that incorporates both estimation and detection performance ('BAYES_full') in order to fully account for the task behavior. This time, the acquired priors consisted of both the expectations about the direction of stimuli motion (θ) and the expectations about whether stimulus is presented (s = 1) or not (s = 0). It was parameterized as:

$$p_{exp}(\theta, s) = \begin{cases} (1-b) \cdot \frac{1}{2\pi}, & \text{if } s = 0 \\ b \cdot \frac{1}{2}[V(-\theta_{exp}, \kappa_{exp}) + V(\theta_{exp}, \kappa_{exp})], & \text{if } s = 1 \end{cases}$$

where parameter b accounts for a participant's average expectation that the stimulus will be presented. Thus, we assumed that expectations about motion direction were uniform for when no stimulus was expected. While the expectations about motion direction when the stimulus was expected followed the bimodal probability distribution just as in the previous models.

On each trial, given the presented motion direction ($\theta_{act}$) and the presence of the stimulus (s), participants made sensory measurements $p_{sens}(\theta_{sens}, s_{sens}|\theta_{act}, s)$. For simplicity, we assumed that the sensory probability of whether the stimulus was present ($p_{sens}(s_{sens}|\theta_{act}, s)$) was independent of the sensory input about the motion direction ($p_{sens}(\theta_{sens}|\theta_{act}, s)$). We further assumed that $s_{sens}$ was independent of the presented motion direction $\theta_{act}$, as informed by 'BAYES_var' model (that allowed the sensory likelihood to vary based on the presented motion direction), which did not produce a better fit. As before, the mean of the motion direction was allowed to fluctuate on trial-by-trial basis, such that:

$$p(\theta|\theta_{act}) = V(\theta_{act}, \kappa_{sens}), \tag{7}$$

where $\kappa_{sens}$ is sensory precision. Given the estimate of the mean θ, the sensory input $\theta_{sens}$ is represented with the associated uncertainty via:

$$p_{sens}(\theta_{sens}|\theta) = V(\theta, \kappa_{sens}). \tag{8}$$

Putting all this together, the sensory likelihood was expressed as:

$$p_{sens}(\theta_{sens}, s_{sens}|\theta, s) = p_{sens}(\theta_{sens}|\theta, s)p(s_{sens}|s) \tag{9}$$

where $p_{sens}(\theta_{sens}|\theta_{act}, s)$ was parameterized as:

$$p_{secs}(\theta_{sens}|\theta_{act}, s) = \begin{cases} \frac{1}{2\pi}, & \text{if } s = 0 \\ V(\theta, \kappa_{sens}), & \text{if } s = 1 \end{cases}$$

where we assumed that sensory likelihood is uniform when no stimulus is presented. Finally, $p_{sens}(s_{sens}|s)$ was parameterized as:

$$p_{sens}(s_{sens} = \{0,1\}|s) = \begin{cases} \{1-c, c\}, & \text{if } s = 0 \\ \{1-d, d\}, & \text{if } s = 1 \end{cases}$$

where parameter c is the average probability of detecting dots when they are not presented, and parameter d is the average probability of detecting dots when they are presented. Putting together prior and likelihood, the resulting posterior probability distribution becomes:

$$p_{post}(\theta, s|\theta_{sens}, s_{sens}) \alpha \, p_{sens}(\theta_{sens}|\theta, s) \cdot p_{sens}(s_{sens}|s) \cdot p_{exp}(\theta, s), \tag{10}$$

With a given posterior participants could have performed detection task at least in two ways. One way is to maximize the posterior (i.e. to always choose the value of s that has higher probability):

$$s_{perc} = \mathrm{argmax}_s\left[p_{post}(s|\theta_{sens}, s_{sens})\right] \tag{11}$$

Another way is to perform probability matching and choose in accordance to the size of the probabilities:

$$s_{perc} = \begin{cases} 0, & \text{if } p_{post}(s=0|\theta_{sens}, s_{sens}) > \eta \\ 1, & \text{if } p_{post}(s=0|\theta_{sens}, s_{sens}) < \eta \end{cases}$$

where $\eta \in [0\ 1]$ and is drawn for each trial from a uniform distribution. We considered both of these possibilities and implemented a variant of the model for each. Finally, just as in 'BAYES' model, the motion direction percept was formed by taking the mean of the posterior:

$$\theta_{perc} = \int \theta . p_{post}(\theta|\theta_{sens}, s_{sens}) d\theta = \frac{1}{Z}\int \theta . \sum_s p_{exp}(\theta) . p_{sens}(\theta_{sens}|\theta, s) . p_{sens}(s_{sens}|s) d\theta, \tag{12}$$

As previously, we accounted for motor precision and the lapse responses via:

$$p(\theta_{est}|\theta_{perc}) = (1-\alpha) \cdot V(\theta_{perc}, \kappa_{motor}) + \alpha \cdot p_{exp}(\theta) * V(0, \kappa_{motor}). \tag{13}$$

In total, 'BAYES_full' model had seven free parameters. To fit the model, in addition to intermediate contrast trials, we also used no-stimulus trial data. The rest of the fitting procedure was the same as in the main text: we built a distribution of 1000 posterior estimations for each presented angle and one more distribution of 1000 posterior estimations for no stimulus trials.

We found that 'BAYES_full' provided a good fit and captured the main features of both estimation and detection performance (*Appendix 2—figure 2*). As before, to test how reliable parameters estimated for 'BAYES_full' model are, we performed parameter recovery. Just as for 'BAYES' parameter recovery described in the main text, we generated 80 sets of parameters and simulated 200 trials of data with 'BAYES_full' model for each of them. Then we fitted 'BAYES_full' to the simulated data. The results revealed that parameters d and c had very poor recovery (*Appendix 2—figure 3*). We thus focused on the simpler model in the current study.

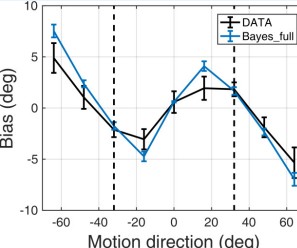 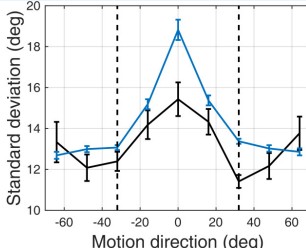 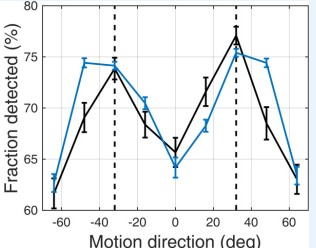

**Appendix 2—figure 2.** Task performance as predicted by the BAYES_full model. Left panel: mean estimation bias at different motion directions. Middle panel: standard deviation of estimations at different motion directions. Right panel: fraction of detected stimuli at different motion directions. The dashed lines correspond to the two most frequently presented motion directions (±32°). Error bars represent within-subject standard error.
DOI: https://doi.org/10.7554/eLife.34115.028

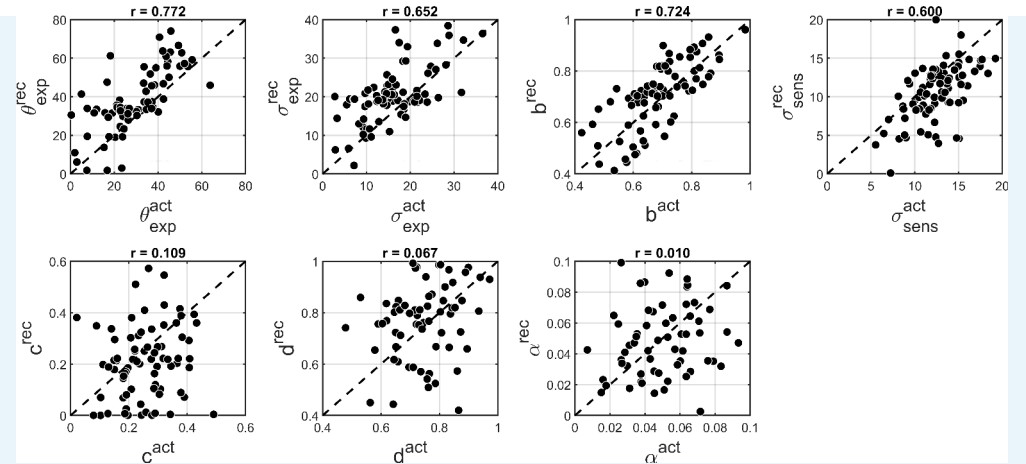

**Appendix 2—figure 3.** Comparison of actual and recovered parameters via 'BAYES_full' model. $\theta_{exp}$ - the mean of prior expectations of motion direction, $\sigma_{exp}$ - uncertainty of the prior expectations of motion direction, $\sigma_{sens}$ - uncertainty in the sensory likelihood, $\alpha$ - fraction of random estimations, $b$ - prior expectation for dots being presented, $c$ likelihood of detecting the dots when they are not presented, $d$ - likelihood of detecting the dots when they are presented. Actual parameters are scattered along x-axis and recovered parameters are scattered along y-axis. The dashed diagonal line is a reference line indicating perfect parameter recovery.

DOI: https://doi.org/10.7554/eLife.34115.029

