## [Decision Letter]

Thank you for submitting your article "Autistic traits, but not schizotypy, predict overweighting of sensory information in Bayesian visual integration" for consideration by *eLife*. Your article has been reviewed by 3 peer reviewers, one of whom, Klaas Enno Stephan (Reviewer #1), is a member of our Board of Reviewing Editors, and the evaluation has been overseen by a Reviewing Editor and Michael Frank as the Senior Editor. The following individuals involved in review of your submission have agreed to reveal their identity; Albert Powers (Reviewer #2); Rick Adams (Reviewer #3).

The reviewers have discussed the reviews with one another and the Reviewing Editor has drafted this decision to help you prepare a revised submission.

Summary:

In this carefully conducted study, the authors use a statistical learning paradigm to quantify the contributions of likelihoods and priors to visual perception in healthy participants with different schizotypal and autistic traits. The authors demonstrate an increased precision of sensory information compared to prior beliefs in participants with autistic traits, but do not find alterations in the precision of sensory information or priors in those with schizotypal traits. While all reviewers found this a compelling and important study, there are several issues that would need to be considered in a revision of the paper.

Essential revisions:

1) The study does not examine patients but exploits variability in questionnaire-based constructs of personality traits (autistic and schizotypal traits) in the healthy population. These are often thought of as lying on a joint spectrum with the clinical conditions (ASD, schizophrenia). While not unreasonable, all reviewers expressed concern about the implicit message of the paper that its findings will necessarily extend to clinical populations. So far, there is no firm empirical evidence that the spectrum covered by AQ and RISC/SPQ, respectively, directly connects to the spectrum of patients that fulfill DSM/ICD diagnoses. The reader should not be given the impression that the results presented here will necessarily and without doubt extrapolate to the clinical conditions. Whether or not the current results extend to clinical populations will still have to be examined in the future; this is something worth emphasising in the Abstract, the Introduction and the Discussion.

2) The analysis deploys a large number of null hypothesis tests but no correction for multiple testing. It would be helpful if, for each set of tests concerning the same question (e.g. Figure 4, 7), you reported corrected p-values, perhaps alongside "native" p-values.

3) The regression plots in Figures 4, 7 show some prominent outliers. It may be worth considering "robust regression" techniques?

4) The negative results for schizotypal traits are described as an absence of effect or relation. This is a little problematic in a frequentist setting (given that one cannot accept the null hypothesis). It would be conceptually cleaner and more interpretable if you computed the Bayes factor for the null hypothesis vs. the alternative hypothesis. A similar issue arises in the subsection “No-stimulus trials and autistic/schizotypal traits” (also discussed in the subsection “Model parameters and autistic/schizotypal traits”) – "suggesting no differences in the acquired prior distributions". As an alternative to a Bayesian analysis, you could state what effect sizes you were powered to reliably detect. This also pertains to the Discussion, and the degree to which this study might contradict other studies.

5) Figure 6D: do you have an explanation why the maxima of the estimated prior probabilities are different from +/- 32 degrees? Is this a reason for concern?

6) Because of the importance of specific symptom dimensions in the literature, it is difficult to reconcile the results at hand with the extant literature without: a) full reporting of SPQ and RISC scores in that group; and b) specific data on the dimensions of schizotypy most present in this sample. A division of scores based upon a factor analysis or relevance to hallucinations would be particularly helpful. Similarly, a breakdown of SPQ based upon symptom dimensions (especially those most relevant to perception) could be enlightening.

7) Similarly, while it is true that the authors focus mostly on ASD in this paper, it would be nice to see data (broken down similar to Figure 3) for the schizotypal group. This would allow the reader to compare performance across groups, which is critical for interpretation.

8) The authors' idea of testing learning rate in autism is a good one, and the authors are correct in stating that their findings support the hypothesis that there may be an increased learning rate in ASD. However, this is not consistent with one work the authors cite (Lawson et al., 2017). Lawson and colleagues find increased volatility estimates in ASD, driving decreased learning. The authors should attempt to reconcile this finding with theirs and perhaps temper their stated expectations for future investigations into learning rate in ASD.

9) An important contrast with the Teufel et al. study is that they used the CAPS and PDI to assess schizotypal hallucinatory and delusional propensity respectively. They found that the CAPS explained all the relation to increased prior 'precision' – i.e. there was no correlation with PDI once CAPS score was partialed out. This implies that greater prior 'precision' is not really related to schizotypy per se but to the subconstruct of aberrant perception. In the 74 item SPQ, only 9 questions refer to perceptual experiences (one could debate whether ideas of reference are also relevant as they may begin with vivid percepts). I'm not familiar with the RISC but I imagine not many of its questions are perceptual. So it seems doubtful that you can really conclude much about your findings in relation to the Teufel et al. study, given such different measures were used. Please mention this in the Discussion. It would be interesting to know whether you detect effects if you only use RISC/SPQ scores relating to perceptual phenomena.

---

## [Author Response]

Essential revisions:

1) The study does not examine patients but exploits variability in questionnaire-based constructs of personality traits (autistic and schizotypal traits) in the healthy population. These are often thought of as lying on a joint spectrum with the clinical conditions (ASD, schizophrenia). While not unreasonable, all reviewers expressed concern about the implicit message of the paper that its findings will necessarily extend to clinical populations. So far, there is no firm empirical evidence that the spectrum covered by AQ and RISC/SPQ, respectively, directly connects to the spectrum of patients that fulfill DSM/ICD diagnoses. The reader should not be given the impression that the results presented here will necessarily and without doubt extrapolate to the clinical conditions. Whether or not the current results extend to clinical populations will still have to be examined in the future; this is something worth emphasising in the Abstract, the Introduction and the Discussion.

We agree with the reviewers that the question of whether or not the current results extend to clinical populations is an open one. We have modified the Introduction, the Discussion and the Conclusion to make this point clearer.

2) The analysis deploys a large number of null hypothesis tests but no correction for multiple testing. It would be helpful if, for each set of tests concerning the same question (e.g. Figure 4, 7), you reported corrected p-values, perhaps alongside "native" p-values.

We now include Bonferroni correction for multiple comparisons and report it alongside the native values where appropriate. The native values now correspond to robust regression rather than to rank correlations (see the response to the essential revision 3). The Bonferroni correction itself was based on the number of independent hypotheses that we tested; that is, we were interested whether ASD and schizotypy had any effect on acquired priors or sensory likelihoods, which resulted in 4 different hypotheses. Even though the number of the null hypothesis significance tests that we performed was higher, the tests all concerning a particular hypothesis were highly dependent (e.g., behavioral biases/variability and priors/likelihoods as recovered via modelling are directly causally related; RISC and SPQ both aim to measure schizotypy construct). Including each individual test as independent would have resulted in overcorrection of p-values, as Bonferroni rests on the assumption that the tests are independent. We include these details in the Statistical tests section in the Materials and methods. The main result of our study, the negative relationship between AQ and sensory uncertainty, remained significant after applying the correction. Furthermore, decreased behavioral variability and decreased number of hallucinations in ASD also remained significant. The only relationship that did not survive the correction was the one between AQ and estimation bias.

3) The regression plots in Figures 4, 7 show some prominent outliers. It may be worth considering "robust regression" techniques?

We have re-run the analysis using robust regression instead of rank correlations. This was done using *robustfit* function in Matlab, which downweighs the influence of outliers via iteratively reweighted least squares (IRLS). For the loss function we used Huber function with a tuning constant of 1.345, which corresponds to 95% estimator efficiency as compared to ordinary least squares. We include these details in the Statistical tests section in the Materials and methods. The results are for the most part consistent with the previous Spearman’s rank correlation analysis. The only difference that is worth noting is the relationship between estimation bias and AQ, which went from significant to trending (i.e., from rho = -0.228, p = 0.039 to r = -0.175, p = 0.053).

4) The negative results for schizotypal traits are described as an absence of effect or relation. This is a little problematic in a frequentist setting (given that one cannot accept the null hypothesis). It would be conceptually cleaner and more interpretable if you computed the Bayes factor for the null hypothesis vs. the alternative hypothesis. A similar issue arises in the subsection “No-stimulus trials and autistic/schizotypal traits” (also discussed in the subsection “Model parameters and autistic/schizotypal traits”) – "suggesting no differences in the acquired prior distributions". As an alternative to a Bayesian analysis, you could state what effect sizes you were powered to reliably detect. This also pertains to the Discussion, and the degree to which this study might contradict other studies.

We thank the reviewers for bringing this up. We now report Bayes Factors for the null hypothesis (BF01) that schizotypy and autistic traits have no effect on the acquired priors. The Bayesian analysis was done using JASP (Version 0.8.6). BF01 were computed for the non-parametric Kendall’s Tau-b correlation coefficient, which maintains a degree of robustness (as compared to Pearson’s correlation – the only other alternative available in JASP). We provide all the details to the reader in the Statistical tests section in the Materials and methods.

5) Figure 6D: do you have an explanation why the maxima of the estimated prior probabilities are different from +/- 32 degrees? Is this a reason for concern?

This phenomenon is indeed intriguing and, aside from the inter-individual variability possibly not completely averaging out, we do not have a good explanation for it. One possibility is that this may be due to how the circular space is sampled around the most frequent directions: there is only a narrow 32 degrees range (from 0 to 32 in between the two frequent directions) where estimates would shift the average “inwards”, in the opposite direction than we currently observe. Whereas estimates following in the remaining – much larger – 32 to 180 range would be leading to “outwards” biases, like what we see now. If the inter-individual variability is sufficiently large, such asymmetry could result in less individuals’ estimations within the 0 to 32 range then within the remaining 32 to 180 range, shifting the group mean accordingly. This phenomenon has been consistently present in previous studies using this task, including the original study by Chalk et al. (2010). It is also worth pointing out that the modes being shifted away from +/- 32 are consistent with the estimation biases in the raw behavioral data, which are also non-zero at +/- 32, therefore it is not a parameter estimation issue.

6) Because of the importance of specific symptom dimensions in the literature, it is difficult to reconcile the results at hand with the extant literature without: a) full reporting of SPQ and RISC scores in that group; and b) specific data on the dimensions of schizotypy most present in this sample. A division of scores based upon a factor analysis or relevance to hallucinations would be particularly helpful. Similarly, a breakdown of SPQ based upon symptom dimensions (especially those most relevant to perception) could be enlightening.

The distribution (mean, standard deviation and range) of SPQ and RISC scores in our sample is reported in ‘Task performance and autistic/schizotypy traits’ section in Results. We agree that a breakdown of SPQ based upon symptom dimensions could be interesting. Unfortunately, we are not able to perform this post-hoc analysis at this time as, due to an unfortunate clerical error, the sub-scores of each dimension of SPQ have not been kept on record (only the total score). RISC however does not include any dimensions that would be considered primarily perceptual (it consists of dimensions of suspiciousness, magical ideation, ritualistic thinking, thought isolation and delusional ideation). Whether inconsistencies between our findings and those reported by previous studies might be underlied by differences in the aberrant perception dimension in the studied groups will be studied in future work.

7) Similarly, while it is true that the authors focus mostly on ASD in this paper, it would be nice to see data (broken down similar to Figure 3) for the schizotypal group. This would allow the reader to compare performance across groups, which is critical for interpretation.

We have now included such figures for both RISC and SPQ scores in the Appendix 1 (Figures 6 and 7).

8) The authors' idea of testing learning rate in autism is a good one, and the authors are correct in stating that their findings support the hypothesis that there may be an increased learning rate in ASD. However, this is not consistent with one work the authors cite (Lawson et al., 2017). Lawson and colleagues find increased volatility estimates in ASD, driving decreased learning. The authors should attempt to reconcile this finding with theirs and perhaps temper their stated expectations for future investigations into learning rate in ASD.

We thank the reviewers for pointing this out. We have included a short discussion about the apparent inconsistency between our findings and those of Lawson et al. (2017). With regards to this, the most relevant difference between our paradigm and theirs, is that Lawson et al. (2017) introduce several (hierarchical) levels of uncertainty and their main results concern an effect that we don’t investigate (our environment being static): the *change* in the learning rates between stable and volatile conditions, rather than the learning rate themselves. Interestingly, for the average learning rates (which we speculate to be increased in ASD), they find no effect.

9) An important contrast with the Teufel et al. study is that they used the CAPS and PDI to assess schizotypal hallucinatory and delusional propensity respectively. They found that the CAPS explained all the relation to increased prior 'precision' – i.e. there was no correlation with PDI once CAPS score was partialed out. This implies that greater prior 'precision' is not really related to schizotypy per se but to the subconstruct of aberrant perception. In the 74 item SPQ, only 9 questions refer to perceptual experiences (one could debate whether ideas of reference are also relevant as they may begin with vivid percepts). I'm not familiar with the RISC but I imagine not many of its questions are perceptual. So it seems doubtful that you can really conclude much about your findings in relation to the Teufel et al. study, given such different measures were used. Please mention this in the Discussion. It would be interesting to know whether you detect effects if you only use RISC/SPQ scores relating to perceptual phenomena.

We thank the reviewers for sharing this clarification. Indeed, very few items in RISC could be construed to be related to aberrant perception, as the questionnaire does not explicitly include such dimension (it consists of dimensions of suspiciousness, magical ideation, ritualistic thinking, thought isolation and delusional ideation). Thus, the subconstruct of aberrant perception could indeed potentially explain the differences in findings. We have revised the Discussion accordingly.